# Structural basis for VPS34 kinase activation by Rab1 and Rab5 on membranes

Shirley Tremel [1], Yohei Ohashi [1], Dustin R. Morado[1,2], Jessie Bertram[1], Olga Perisic [1], Laura T. L. Brandt [1], Marie-Kristin von Wrisberg [3], Zhuo A. Chen [4], Sarah L. Maslen [1], Oleksiy Kovtun [1], Mark Skehel [1], Juri Rappsilber[4,5], Kathrin Lang [3], Sean Munro [1✉], John A. G. Briggs [1✉] & Roger L. Williams [1✉]

The lipid phosphatidylinositol-3-phosphate (PI3P) is a regulator of two fundamental but distinct cellular processes, endocytosis and autophagy, so its generation needs to be under precise temporal and spatial control. PI3P is generated by two complexes that both contain the lipid kinase VPS34: complex II on endosomes (VPS34/VPS15/Beclin 1/UVRAG), and complex I on autophagosomes (VPS34/VPS15/Beclin 1/ATG14L). The endosomal GTPase Rab5 binds complex II, but the mechanism of VPS34 activation by Rab5 has remained elusive, and no GTPase is known to bind complex I. Here we show that Rab5a–GTP recruits endocytic complex II to membranes and activates it by binding between the VPS34 C2 and VPS15 WD40 domains. Electron cryotomography of complex II on Rab5a-decorated vesicles shows that the VPS34 kinase domain is released from inhibition by VPS15 and hovers over the lipid bilayer, poised for catalysis. We also show that the GTPase Rab1a, which is known to be involved in autophagy, recruits and activates the autophagy-specific complex I, but not complex II. Both Rabs bind to the same VPS34 interface but in a manner unique for each. These findings reveal how VPS34 complexes are activated on membranes by specific Rab GTPases and how they are recruited to unique cellular locations.

[1] MRC Laboratory of Molecular Biology, Cambridge, UK. [2] Science for Life Laboratory, Department of Biochemistry and Biophysics, Stockholm University, Solna, Sweden. [3] Center for Integrated Protein Science Munich (CIPSM), Department of Chemistry, Lab for Synthetic Biochemistry, Technical University of Munich, Institute for Advanced Study, TUM-IAS, Garching, Germany. [4] Bioanalytics, Institute of Biotechnology, Technische Universität Berlin, Berlin, Germany. [5] Wellcome Centre for Cell Biology, University of Edinburgh, Edinburgh, UK. ✉email: sean@mrc-lmb.cam.ac.uk; jbriggs@mrc-lmb.cam.ac.uk; rlw@mrc-lmb.cam.ac.uk

VPS34 is the primordial member of the phosphoinositide 3-kinase (PI3K) family of lipid kinases, and it is present in all eukaryotic clades. It phosphorylates phosphatidylinositol (PI) to generate PI3P, which controls a range of cellular processes by reversibly recruiting specific protein effectors to membranes. Stable membrane association of effector proteins, as well as the VPS34 complexes themselves, is often achieved by a combination of low-affinity membrane interactions and an additional determinant, such as a small GTPase[1]. The Rab family of small GTPases are fundamental to intracellular membrane trafficking in which organelles exchange proteins and lipids in an intricate network of vesicular transport. Furthermore, many viral and bacterial pathogens exploit Rabs for entry and survival[2–6], and Rab misregulation is associated with human diseases such as neurodegeneration and cancer[7–11]. Rab5 is indispensable at early endosomes[12], where it has a role in the activation of VPS34[13]. PI3P positively regulates the activity of Rab5 by recruiting it to membranes through a positive feedback loop with its GEF/effector complex, Rabex5/Rabaptin5[14]. Human VPS34 forms two heterotetrameric core complexes known as complexes I and II[15]. Complex I is composed of VPS34, VPS15, Beclin 1, and ATG14L, whereas complex II has UVRAG instead of ATG14L. This difference in a single subunit determines how and where the two complexes are active. Complex I produces PI3P at the phagophore, promoting autophagosome formation, while complex II has a main role in endocytic sorting[15–17], along with other cellular pathways[18–20]. Complexes I and II uniquely respond to the nature of the lipid membranes in which they find their substrates[21]. However, the broader interaction of the entire complex with membranes and membrane-associated proteins remains unclear. Furthermore, although VPS34 and VPS15 were identified as Rab5 interactors, the mechanism of activation has remained elusive for over 20 years[13,22].

Here we show that Rab5a recruits and profoundly activates endocytic complex II on membranes, using unnatural amino acid (UAA) mediated cross-linking, hydrogen/deuterium exchange mass spectrometry (HDX-MS), electron cryotomography (cryo-ET), and a reconstituted kinase assay. Our cryo-ET structure of complex II bound to Rab5a-decorated membranes shows how the complex directly engages with the membrane and how it is activated by Rab5a. We also find that the GTPase Rab1a is an exclusive activator of the autophagy-specific VPS34 complex I but not complex II.

## Results

**Rab5a is a potent activator of complex II.** We first examined the effect of membrane-anchored Rab5a on the lipid kinase activity of human VPS34 complexes on giant unilamellar vesicles (GUVs). In cells, Rab5 is C-terminally prenylated, and this modification anchors the GTPase on endosomal membranes[23]. In order to mimic membrane attachment, Rab5a (Q79L), loaded with either GDP or GTP, was coupled to GUVs through a covalent bond between the C-terminal cysteine and a maleimide-conjugated lipid in the membrane (Supplementary Fig. 1a). The immobilised Rab5a potently activated complex II up to 40-fold in a GTP-dependent manner (Fig. 1a), but activated complex I only modestly by threefold (Fig. 1b). Soluble Rab5a–GTP had no influence on the activity of VPS34 complexes (Supplementary Fig. 1b). Rab5a-GDP coupled to GUVs showed only a modest activation of complex II (threefold) (Fig. 1a) and no activation of complex I (Fig. 1b). Vesicle flotation assays showed that 100 nm large unilamellar vesicles (LUVs) containing membrane-anchored Rab5a–GTP efficiently recruited complex II to membranes in a GTP-dependent manner (Fig. 1c and Supplementary Fig. 2a) but less-efficiently recruited complex I (Supplementary Fig. 2b).

Taken together, these results show that recruitment of VPS34 complex II to membranes by Rab5a is GTP-dependent and essential for the activation of the complex.

**Mapping the Rab5a–GTP binding site on complex II.** To map the Rab5a–GTP interaction site on complex II, we first employed crosslinking by genetic code expansion. Rab5a (Q79L) with an amber mutation was coexpressed with the bromoalkyl-bearing unnatural amino acid BrCO6K and an orthogonal pyrrolysyl-tRNA synthetase/tRNA pair in bacteria (Fig. 1d)[24]. BrCO6K is a lysine derivative that can crosslink to proximal nucleophilic amino acids such as cysteines, aspartates and glutamates to bridge distances up to 15 Å under physiological conditions[24,25]. A Rab5a mutant, in which the codon for S84 in switch 2 was replaced by the amber codon, showed the most efficient crosslinking (Fig. 1d). Both complex II and VPS34 alone incubated with Rab5a–GTP–BrCO6K gave rise to a crosslinked product that ran slightly above the 120 kDa molecular weight marker, consistent with 102 kDa VPS34 crosslinked to the 24 kDa Rab5a (126 kDa). Rab5a crosslinked much more efficiently to complex II than to VPS34 alone, suggesting that VPS15, Beclin 1 or UVRAG also influence the Rab5a–GTP interaction. Analysis by mass spectrometry of the crosslinked products identified Rab5a residue 84 crosslinked to VPS34 E202 (Supplementary Table 1). Residue E202 is near the C-terminal end of helix α2 in the helical hairpin insertion (C2HH) in the VPS34 C2 domain (Fig. 1e).

The crosslinking results were further confirmed by HDX-MS of complex II in the presence and absence of Rab5a (Q79L) loaded with GTP (Fig. 1e and Supplementary Fig. 3a, Supplementary Data 1). A stretch of helix α2 in the VPS34 C2HH insertion (HDX 204–206) showed a decrease in HDX. Furthermore, a reduction in deuterium incorporation was seen in VPS15 in the WD40 domain (HDX 1213–1224 and 1278–1299) and in a small globular domain in VPS15 (HDX 771–787) that we will refer to as the VPS15 SGD (the domain consisting of residues 771–816). Combined with the observed BrCO6K crosslinks, we could make a model for the Rab5a–GTP interaction in which Rab5a binds to a tripartite binding site made of the VPS34 C2HH insertion, the VPS15 SGD domain and the WD40 domain (Fig. 1e and Supplementary Fig. 3a).

In order to validate the importance of the observed VPS34/Rab5a interface, we assayed complex II recruitment by Rab5a in cells. mCherry-tagged Rab5a-Q79L transfected into HEK293T cells caused Rab5a-positive enlarged endosomes as expected (Fig. 2a–d)[26,27]. VPS34-EGFP cotransfected along with the other complex II components (VPS15, Beclin 1, UVRAG), colocalized with mCherry–Rab5a–Q79L (Fig. 2a, e). We next tested the importance of the C2HH in VPS34 for binding Rab5a by mutating C2HH residues 199-REIE-202 to alanine (REIE > AAAA). Surprisingly, Rab5a colocalization was markedly increased with complex II carrying VPS34 REIE > AAAA (Fig. 2b, e). No colocalization with Rab5a was observed with complex I when VPS34-EGFP (WT or REIE > AAAA) was cotransfected along with VPS15, Beclin 1, and ATG14L (Fig. 2c–e). This mutation has a similar effect in vitro, with complex II REIE > AAAA being activated by Rab5a–GTP fourfold more than the wild-type (Fig. 2f, g).

**Rab1 interacts with and activates the autophagy-associated complex I.** While Rab5 is a long-known effector of VPS34, we wanted to know whether there are other Rab GTPases that directly interact with and regulate VPS34 complexes. We recently carried out a MitoID analysis, which uses proximity biotinylation to identify effectors bound to a mitochondrially-localised form of 11 human Rab GTPases[28]. As expected, we found VPS34 and VPS15

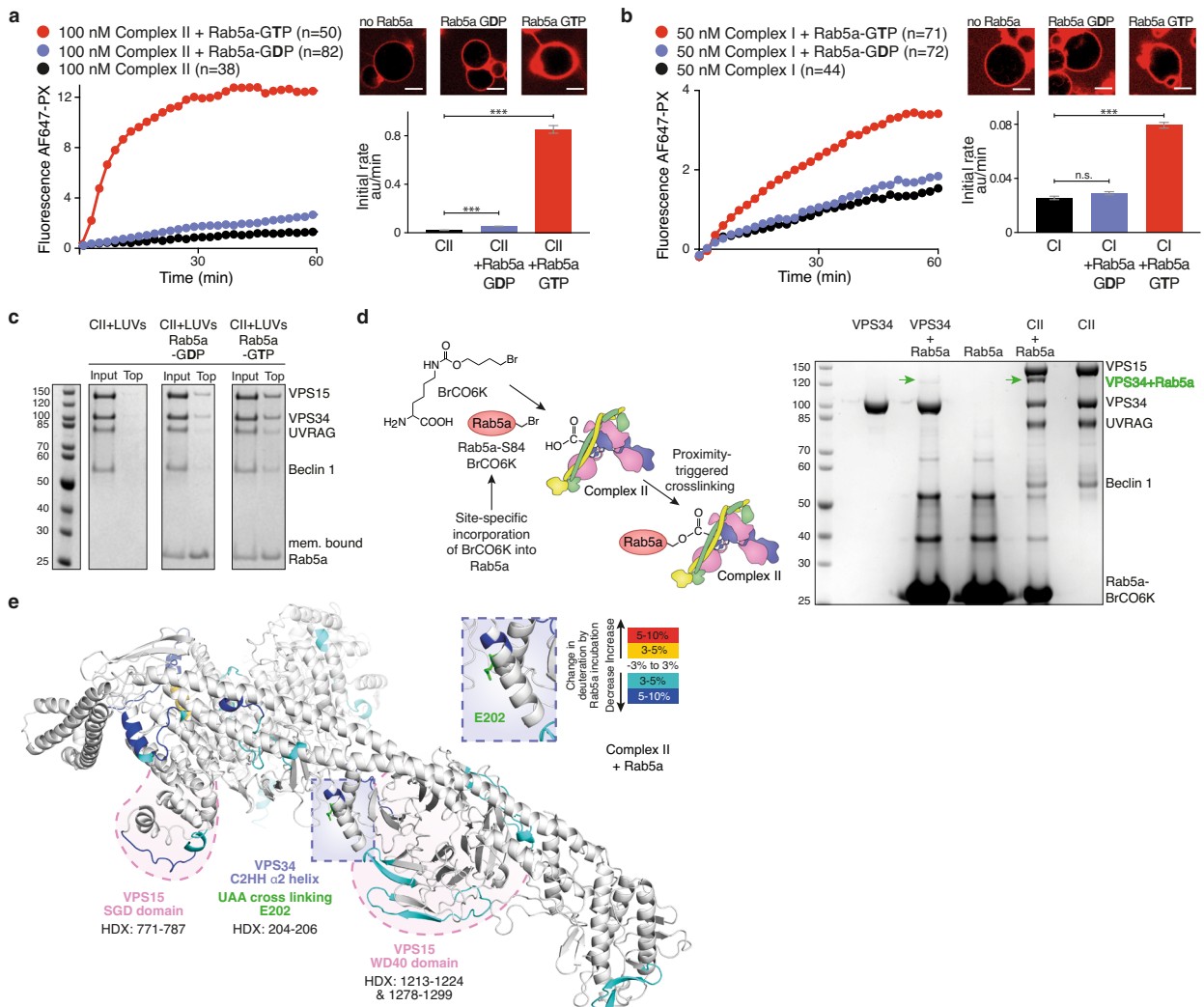

**Fig. 1 Membrane-bound Rab5a–GTP recruits and activates VPS34 complex II. a, b** GUV-based activity with membrane-tethered Rab5a. The reaction progress curves and initial rates are shown. Micrographs: AF647-PX signals at the end of reactions. Scale bars: 5 μm. Bar graphs: initial rates of the reaction curves (AF647-PX fluorescence change/min in arbitrary units, AU). **a** Complex II is greatly activated by membrane-tethered Rab5a in a GTP-dependent manner. **b** Complex I is only modestly activated by membrane-attached Rab5a–GTP. **c** Complex II is recruited to Rab5a-decorated LUVs in a GTP-dependent manner as measured by a flotation assay. Here, complex II is mixed with LUVs and the mixture is added on top of a sucrose gradient and centrifuged. Membrane-bound proteins float up to the top of the gradient as seen in gel lanes (Top). Gel quantification can be found in Supplementary Fig. 2a. **d** Mapping the Rab5a–GTP binding site by proximity-triggered crosslinking of complex II. SDS-PAGE gel of crosslinking reactions, products indicated by green arrows. **e** Mapping the Rab5a binding site on complex II by the HDX-MS. HDX changes are displayed on a model of human complex II created with PyMOL. Rab5a binding protects (coloured in cyan and blue) the VPS34 C2 helical hairpin insertion (C2HH) and the VPS15 SGD and WD40 domains. The VPS34 C2 (E202) that is crosslinked by unnatural amino acid Rab5a-84BrCO6K is coloured green and shown in an expanded panel. Source data are provided as a Source Data file.

amongst the top Rab5-interacting proteins. Here, we expand the MitoID analysis to include Rab1a. Surprisingly, among the identified proteins, we detected all four subunits of VPS34 complex I, but not the UVRAG subunit characteristic for complex II (Fig. 3a). The in vitro kinase assay with Rab1a-decorated GUVs showed that Rab1a–GTP potently activates complex I by 11-fold while Rab1a–GDP only activates by threefold (Fig. 3b). In contrast, membrane-bound Rab1a loaded with either GTP or GDP has no influence on complex II activity (Fig. 3c). Consistent with this, a flotation assay showed that Rab1a-coupled to LUVs efficiently recruited complex I to membranes in a GTP-dependent manner (Fig. 3d and Supplementary Fig. 2c), while it did not recruit complex II (Supplementary Fig. 2d). Differences in HDX between complex I in the presence and absence of Rab1a (Supplementary Data 2) indicated that helix α2 (HDX 194–200 and 204–214) from

the VPS34 C2 domain is in contact with Rab1a (Fig. 3e and Supplementary Fig. 3b), suggesting that Rab1a is likely to bind in the same pocket on complex I as Rab5a on complex II. In contrast to Rab5a-complex II, the VPS15 SGD and the WD40 domain did not show significant HDX reduction, whereas the Beclin 1 coiled-coil domain 2 (CC2, HDX 223–235) shows a significant HDX increase (Fig. 3e).

Consistent with previous reports of Rab1 on the Golgi apparatus[29,30], GTP-locked mCherry-Rab1a–Q70L shows a juxtanuclear localisation (Fig. 4a–d). When wild-type VPS34-EGFP was cotransfected with the other components of complex I, it strongly colocalised with this compartment (Fig. 4a, e). On the other hand, VPS34 complex I carrying the REIE > AAAA mutation showed a markedly reduced colocalization with Rab1a–Q70L (Fig. 4b, e). Furthermore, complex II (VPS34 WT

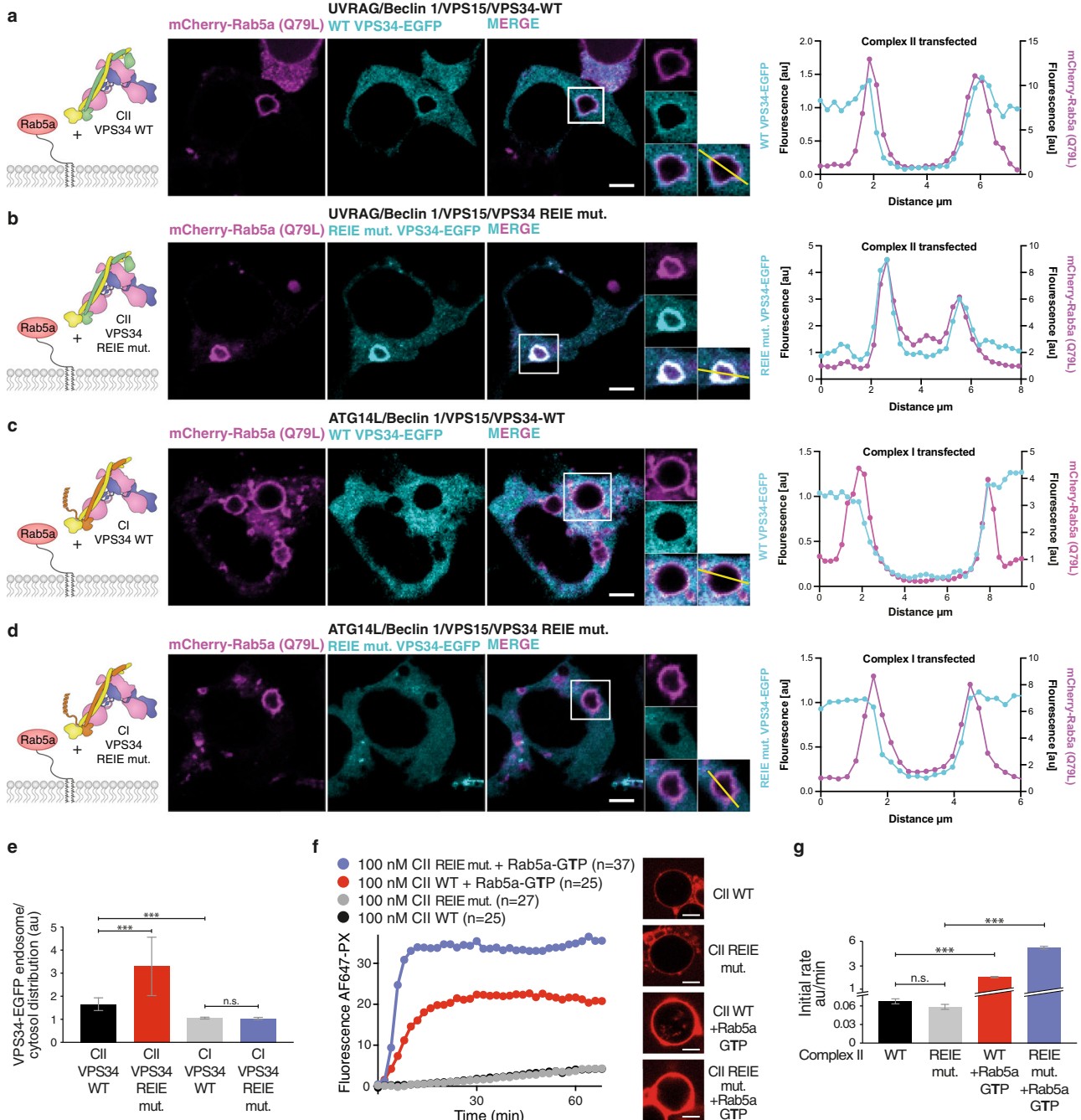

**Fig. 2 The C2 helical insertion (C2HH) of VPS34 is a critical element for the interaction of complex II with Rab5a. a, b** GTP-locked Rab5a (Q79L) and all four components of complex II, containing either C-terminally EGFP-tagged WT (**a**) or C2HH mutant REIE > AAAA VPS34 (**b**), were coexpressed in HEK293T cells. Confocal images show cellular localisation of mCherry–Rab5a-Q79L (magenta) and VPS34-EGFP (cyan). To the right of each panel, traces for the fluorescence along the transect indicated by the line in the micrograph are shown. The C2HH REIE mutant markedly increased colocalization of complex II with Rab5a (**b**, **e**) compared to VPS34 WT (**a**, **e**). **c**, **d** GTP-locked Rab5a (Q79L) and all four components of complex I, containing either C-terminally EGFP-tagged WT (**c**) or mutant REIE > AAAA VPS34 (**d**), were coexpressed in HEK293T cells. The mutant had no impact on the colocalization of complex I with Rab5a (**c–e**), quantitated as described in methods. Error bars: standard deviation. ***: $p < 0.0001$; n.s.: $p > 0.05$. Scale bars: 10 μm. **f** GUV assay of activity of complex II shows that the VPS34 C2HH REIE > AAAA mutation greatly potentiates the activation of complex II by Rab5a–GTP, without affecting the basal activity of complex II. Micrographs: AF647-PX signals at the end of reactions. Scale bars: 5 μm. **g** The initial rates in the GUV assay in **f** (AF647-PX fluorescence change/min in arbitrary units, AU) are depicted. ***: $p < 0.001$; n.s.: $p > 0.05$. Source data are provided as a Source Data file.

or REIE > AAAA) could not be recruited to this compartment (Fig. 4c–e). Consistent with the cellular data, complex I with VPS34 REIE > AAAA mutation was not activated by GUVs coated with Rab1a–GTP, again showing that the liposome activity assay faithfully recapitulates the recruitment of the complexes to membranes in cells (Fig. 4f, g).

**The cryo-ET structure of complex II on Rab5a–GTP-coupled vesicles.** Complexes I and II form a Y-shaped structure: a catalytic arm bearing the kinase domains of VPS34 and VPS15 at its tip and an adaptor arm with Beclin 1 and either UVRAG or ATG14L supporting the VPS15/VPS34 subunits[31–34]. Several structures of complexes I or II have been reported[31–35]. However, none of

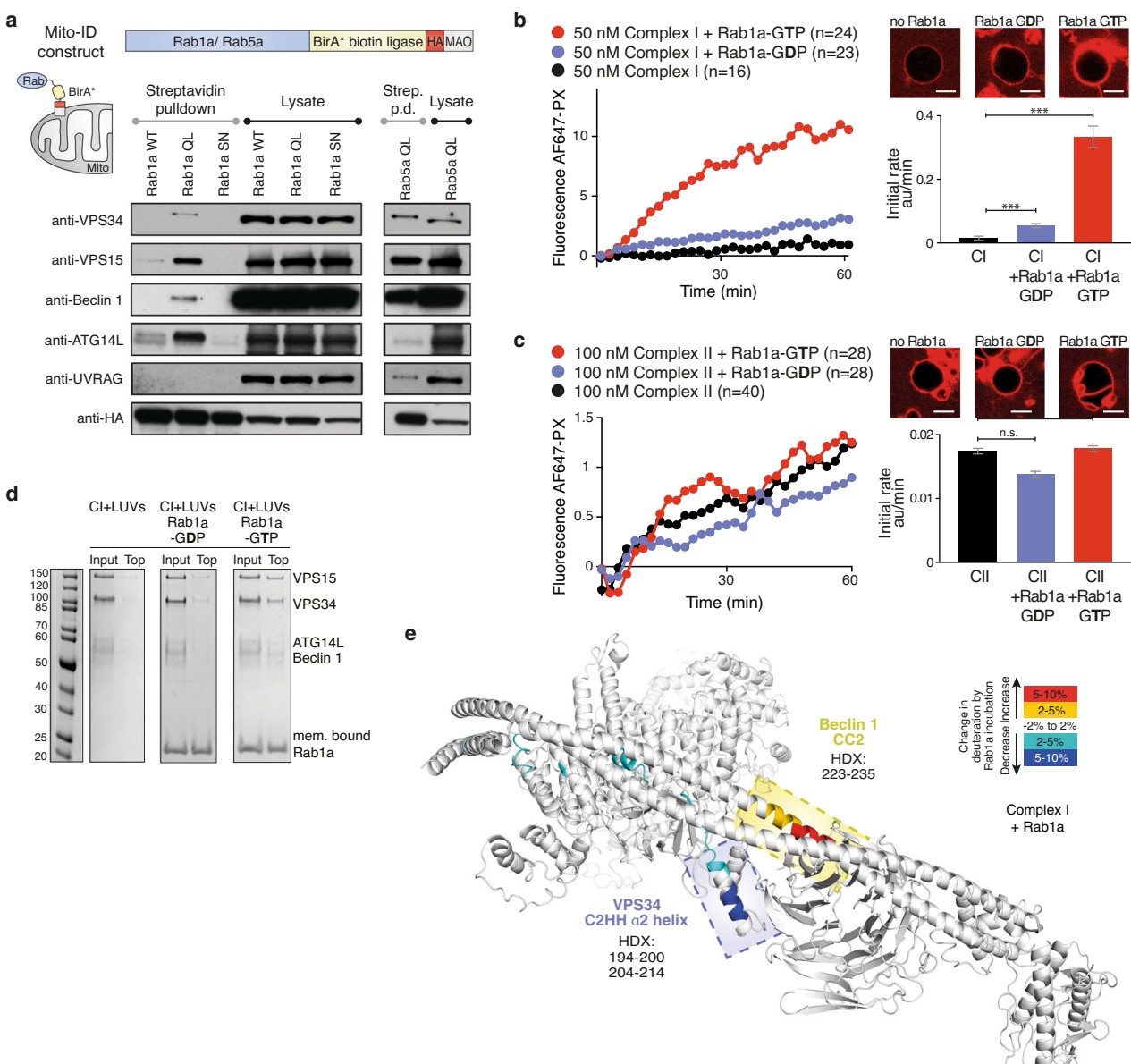

**Fig. 3 Rab1a is a specific activator of VPS34 complex I. a** Immunoblot of streptavidin precipitates from cells in which MitoID had been carried out for Rab1a WT, QL, and SN mutants (left) or Rab5 QL (right). Rab1a interacts specifically with components of complex I (VPS34, VPS15, Beclin 1, ATG14L) but not with the complex II-specific UVRAG. Rab5a shows only a weak interaction with the complex I-specific ATG14L. **b** Complex I was potently activated by membrane-attached Rab1a in a GTP-dependent manner. **c** No activation of complex II by either membrane-attached Rab1a–GTP or Rab1a–GDP was detected. **b**, **c** Micrographs: AF647-PX signals at the end of reactions. Scale bars: 5 μm. Bar graphs: initial rates of the reaction curves. **d** Lipid flotation assays showing complex I recruitment to Rab1a-decorated membranes in a GTP-dependent manner. Gel quantification is in Supplementary Fig. 2c. **e** Mapping Rab1a binding site on complex I by HDX-MS. Rab1a binding increases protection (coloured in cyan and blue) of the VPS34 C2 insertion (C2HH) and decreases protection (coloured yellow and red) of Beclin 1 CC2. Source data are provided as a Source Data file.

these structures directly reveal how these complexes interact with membranes. In order to directly visualise the interaction with membranes, we determined the structure of complex II bound to Rab5a–GTP-decorated LUVs using cryo-ET and subtomogram averaging. To improve the membrane recruitment, we made a construct (complex II–BATS) that has the C terminus of UVRAG deleted (residues 1–464 remaining) and replaced by the BATS domain of ATG14L (residues 413–492)[21,36] which can still be activated by Rab5a–GTP (Supplementary Fig. 4a). We collected 115 tilt series on a Titan Krios electron microscope, of which 105 were used for subtomogram averaging. Subtomograms for averaging were picked from 3008 vesicles with a mean diameter of 81.3 nm and a standard deviation of 19.5 nm (Fig. 5a and

Supplementary Fig. 4b). A total of 26,979 subtomograms contributed to the final average resulting in a reconstruction with an overall resolution of 9.8 Å (Fig. 5b and Supplementary Fig. 5). Several studies of membrane coat proteins such as COP-I/II, retromer or clathrin have been determined using subtomogram averaging[37–40]. Coat proteins assemble into symmetric lattice structures whereas complex II is distributed irregularly over the membrane surface and is randomly rotated.

**Rab5a binds to a tripartite site made of VPS34 and VPS15 on the adaptor arm of complex II.** A model of human complex II based on the yeast complex II crystal structure was built using

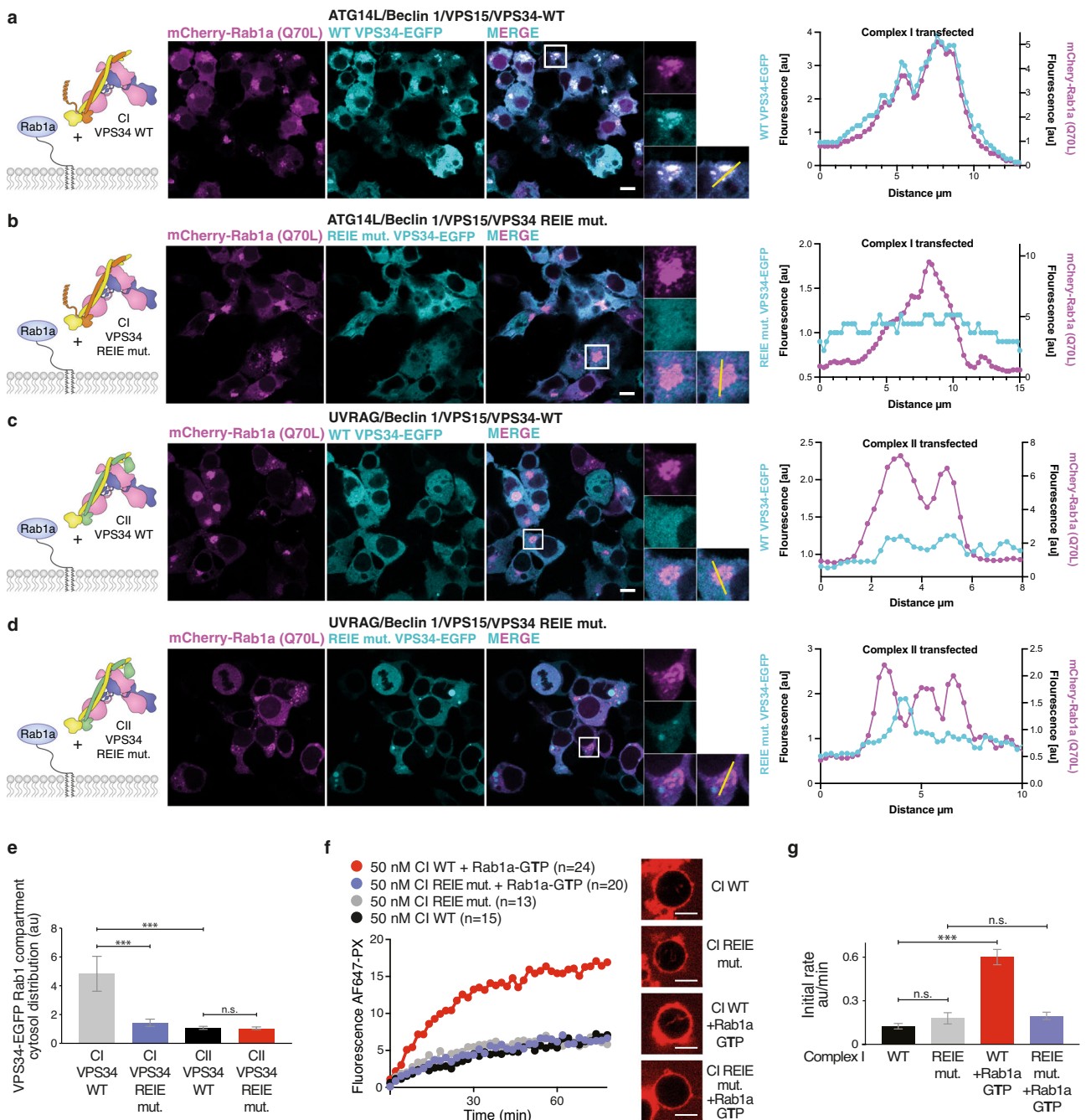

**Fig. 4 The VPS34 C2HH is also a critical element for the interaction of complex I with Rab1a. a, b** GTP-locked Rab1a (Q70L) and all four components of complex I, containing either C-terminally EGFP-tagged WT (**a**) or mutant REIE > AAAA VPS34 (**b**), were coexpressed in HEK293T cells. Confocal images show the localisation of mCherry-Rab1a–Q70L (magenta) and VPS34-GFP (cyan). The REIE mutant in VPS34 reduces colocalization of complex I with Rab1a (**e**, quantitated as described in 'Methods' section). **c, d** GTP-locked Rab1a (Q70L) and all four components of complex II, containing either C-terminally EGFP-tagged WT (**c**) or mutant REIE > AAAA VPS34 (**d**), were coexpressed in HEK293T cells. Complex II shows no significant colocalization with Rab1a for either WT or mutant VPS34 (**e**, quantitated as described in the methods. Error bars: standard deviation. ***: $p < 0.0001$; n.s.: $p > 0.05$). Scale bars: 5 μm. **f** GUV assay of activity of complex I shows that the VPS34 C2HH REIE > AAAA mutation eliminates activation of complex I by Rab1-GTP, without affecting the basal activity. Micrographs: AF647-PX signals at the end of reactions. Scale bars: 5 μm. **g** The initial rates in the GUV assay (AF647-PX fluorescence change/min in arbitrary units, AU) in **f** are depicted. ***: $p < 0.001$; n.s.: $p > 0.05$. Source data are provided as a Source Data file.

SWISSMODEL[41]. This model could generally be fit as a rigid body into the cryo-ET reconstruction, and this was followed by small manual adjustments and refinement with REFMAC[42–45] and ISOLDE[46] (Fig. 5b). Examination of the cryo-ET density revealed a clear extra density at the adaptor arm that could not be allocated to any of the complex II components. The density is located exactly where Rab5a was

positioned by HDX-MS and UAA crosslinking analysis (Fig. 1e) and was of sufficient resolution to unambiguously fit the crystal structure of Rab5a–GMP–PNP (PDB 3MJH[47]) (Fig. 5c and Supplementary Movie 1). Thus, Rab5a–GTP is located in a pocket surrounded by the VPS15 SGD and WD40 domain, the UVRAG/Beclin 1 coiled-coil, and VPS34 C2HH insertion of the C2 domain.

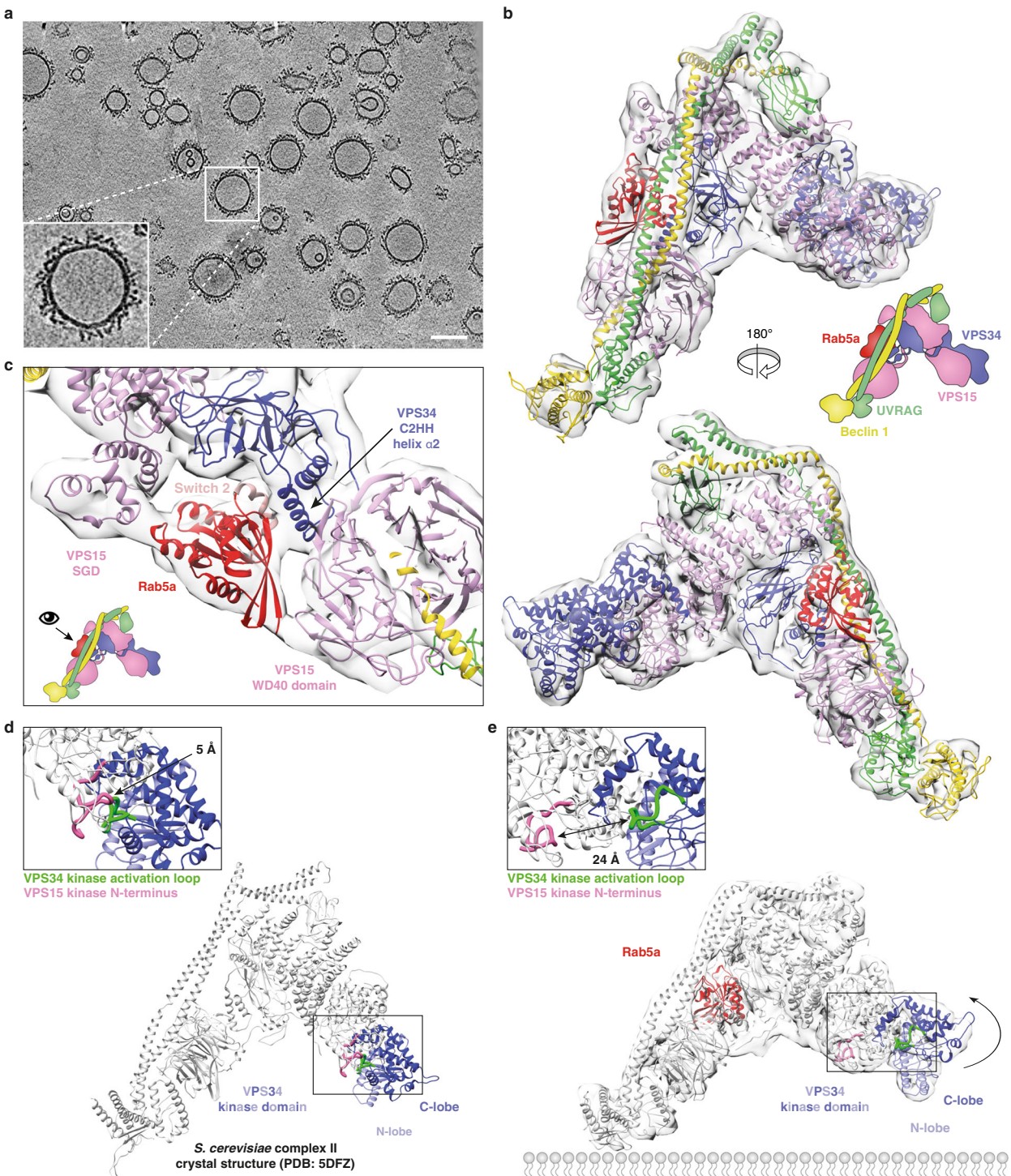

**Fig. 5 Cryo-ET structure of VPS34 complex II on Rab5a–GTP-coated membranes. a** Section through a cryo-electron tomogram of Rab5a-decorated vesicles coated with VPS34 complex II–BATS. Scale bar 100 nm. **b** Subtomogram averaging reconstruction of human complex II–BATS bound to membrane-attached Rab5a–GTP at 9.8 Å with the membrane masked-out. A model of human complex II based on the crystal structure of yeast complex II (5DFZ) was fit to the density and refined with restraints. **c** Rab5a (PDB entry 3MJH) was optimally fit into the cryo-ET density. In this orientation, the Rab5a switch 2 helix faces the VPS34 C2 insertion (C2HH). The Rab5a density is nestled between the C2HH, the VPS15 SGD and WD40 domains. **d** The crystal structure of the inactive state of yeast complex II, in which the VPS15 kinase domain (N terminus coloured pink) contacts and inhibits the VPS34 kinase domain (activation loop coloured green). **e** The cryo-ET structure of the active form of human complex II bound to membrane-attached Rab5a–GTP in the same orientation as **d**.

**Capturing complex II in its activated state**. In the crystal structure of yeast complex II, the Vps34 activation loop and the C-terminal helix, two elements critical for the lipid kinase activity of Vps34, are packed against the N-lobe of the kinase domain of Vps34, and we and others proposed that this represents an

autoinhibited conformation[31,34] (Fig. 5d). As noted above, the cryo-ET density of the catalytic arm could not precisely accommodate the model of the inactive yeast complex II. Instead, it appears that binding to Rab5a-decorated membranes releases the inhibitory contacts between VPS34 and VPS15. The VPS34

HELCAT (helical and kinase domain[48]) is rotated in the cryo-ET structure, released from the inhibitory grasp of the VPS15 kinase domain (Fig. 5e), suggesting allosteric activation of complex II by membrane coupled Rab5a. Similar allosteric activation of complex I by binding NRBF2 was recently reported[34]. This indicates that the activation mechanism of VPS34 complex II by Rab5a involves both the increased recruitment of complex II by Rab5a to membranes and the release of the autoinhibition.

**The BARA domain in the adaptor arm attaches complex II to membranes**. The unmasked cryo-ET structure of complex II–BATS shows a clear density corresponding to the membrane. Interestingly, only the adaptor arm is solidly attached to the membrane while the catalytic arm is poised above the membrane (Fig. 6a, b). Lowering the density threshold shows that the density of the protein complex is continuous with the outer membrane leaflet (Fig. 6c). The BARA domain of Beclin 1 shows a protrusion towards the membrane, which is in direct contact with the lipid bilayer. The Beclin 1 BARA domain contains a hydrophobic loop (aromatic finger 1, residues FFW 359–361), which is critical for membrane binding[21,31,35,49,50]. Thus, the loop was modelled into the protrusion (Fig. 6a–c). Only a short extension at the Beclin 1 BARA domain is visible in the cryo-ET density that could correspond to the ATG14L BATS domain that was fused to UVRAG to increase membrane occupancy (Fig. 6a).

All Rab GTPases have an unstructured C terminus called the hypervariable region (HVR) before the cysteine motif for prenylation and membrane anchoring[51]. The HVR of Rab5a is not resolved, but its length of 34 residues could easily span the distance of ~50 Å observed between the Rab density and the membrane (Fig. 6b).

**Complex II is present in a range of orientations on the lipid bilayer**. We next applied principal component analysis (PCA) classification on wedge-masked difference maps[52] to classify the subtomograms according to the orientation of complex II–BATS relative to the membrane (Supplementary Fig. 4c). Three of these classes in which the complex can adopt a range of orientations with respect to the membrane are shown in Fig. 6d and Supplementary Movie 2. In all classes, the adaptor arm is the anchor point at which the entire complex tilts relative to the membrane and rotates around the adaptor arm. The tilting of the complex could regulate the engagement with membranes by both arms of the complex, while the rotations around the adaptor arm would enable the catalytic arm to survey a larger membrane area for a substrate lipid. This range of movement is mediated by the flexible membrane anchoring of the Beclin 1 BARA domain. Since the mean diameters of the LUVs for these three classes were similar (80.0–83.4 nm) (Supplementary Fig. 4b), the motions of the complex with respect to the membrane are not likely to be caused by different membrane curvatures.

The VPS34 HELCAT has been shown to be able to completely dislodge from complex I, and it was proposed that this movement is essential for maximum activity, enabling the VPS34 HELCAT to engage with PI, which would not be accessible in the classic V-shape[53]. In our cryo-ET, the density for the kinase domain is present in all classes having sharp features in the adaptor arm, suggesting the ordered catalytic arm is a stable feature of the active complex II bound to the Rab5a–GTP-decorated LUVs (Supplementary Fig. 6b). Thus, instead of the HELCAT domain dislodging to access the lipid bilayer, the cryo-ET suggests that the whole complex tilts and rotates to reach the membrane. With its adaptor arm flexibly bound to the lipid bilayer, the complex II is able to explore an extensive range of orientations relative to the membrane in order to bring the VPS34 HELCAT in close proximity to its target PI.

**Discussion**

The cryo-ET structure of human complex II on Rab5a-coupled membranes has revealed the Rab5 binding site on complex II and elucidated key events associated with the activation of VPS34 by Rab5a–GTP. Strikingly, Rab5a-decorated membranes induced allosteric changes in the VPS34 HELCAT that release the VPS34 kinase domain from the inhibition by the VPS15 kinase domain that was observed in the autoinhibited form of the yeast complex II[31] (Fig. 5d, e). This change in conformation of complex II after associating with Rab5a on membranes is consistent with the recent, elegant study of VPS34 complex II by single-molecule kinetics that showed that Rab5a coupled to membranes both increases the density of complex II on membranes and increases the specific activity of complex II on Rab5a membranes relative to membranes without Rab5[54]. While it is clear that a significant component of the increase in activity of complex I on Rab1a-coupled membranes and complex II on Rab5a-coupled membranes is due to enhanced recruitment, the conformational changes in VPS34 seen in the cryo-ET suggest that activation involves multiple steps. The structure shows that the adaptor arm is tightly associated with the lipid bilayer whereas the catalytic arm is poised above it, indicating that the catalytic arm carries out catalysis and then detaches from the membrane (Fig. 6a, b). The reconstruction demonstrates that this poised, active conformation is the prevailing one. This might represent the predominant part of the enzyme's catalytic cycle, or it might have been stalled in this state due to the lack of Mg/ATP in the cryo-ET material. This intermediate conformation could imbue the enzyme complex with properties in-between hopping and scooting catalysis[55] where the scooting adaptor arm is tied to a hopping catalytic arm. The structures suggest a three-state model of complex II activation by Rab5a–GTP (Fig. 6e).

Consistent with the organelle-specific roles of Rab5a on early endosomes[14,56,57] and Rab1 in autophagy[30,58,59], we found that Rab5a preferentially activates complex II over complex I, and Rab1a activates only complex I. The cryo-ET structure shows that the VPS15 WD40 and SGD domains, as well as the VPS34 C2 helical insertion (C2HH), make interactions with Rab5a (Fig. 5c). However, it appears that the most extensive interface with the Rab5a switch regions, which assume nucleotide-specific conformations, is the VPS34 C2HH. The C2HH interacts with both switches I and II, while the VPS15 WD40 domain interacts with Rab5a outside the switches. The VPS15 SGD interacts with the surface of Rab5a opposite the switch interface. Rab1a makes similar interactions with the VPS34 C2 helical insertion in complex I (Fig. 3e). However, the VPS15 WD40 domain is greatly shifted in complex I relative to complex II (Supplementary Fig. 7), making it farther from the putative Rab1a interface, which is consistent with the lack of changes in HDX for the complex I WD40 upon Rab1a binding. This shift of the VPS15 WD40 appears to cause the VPS34 C2HH to tilt as it tracks the movement of the WD40 domain. This might be part of the mechanism for selectivity of complex I for Rab1a and complex II for Rab5a. The complex-specific position of the WD40 domain may be caused by the ATG14L or UVRAG unique domain that packs against the WD40. Alternatively, it may be that there are direct interactions of UVRAG or ATG14L with Rabs that are transient and unobserved in our low-resolution reconstruction from cryo-ET and are part of unobserved regions in HDX-MS analysis for these subunits. Interestingly, the same C2HH mutant that increases activation of complex II by Rab5a markedly reduces activation of complex I by Rab1a, which could provide a tool to

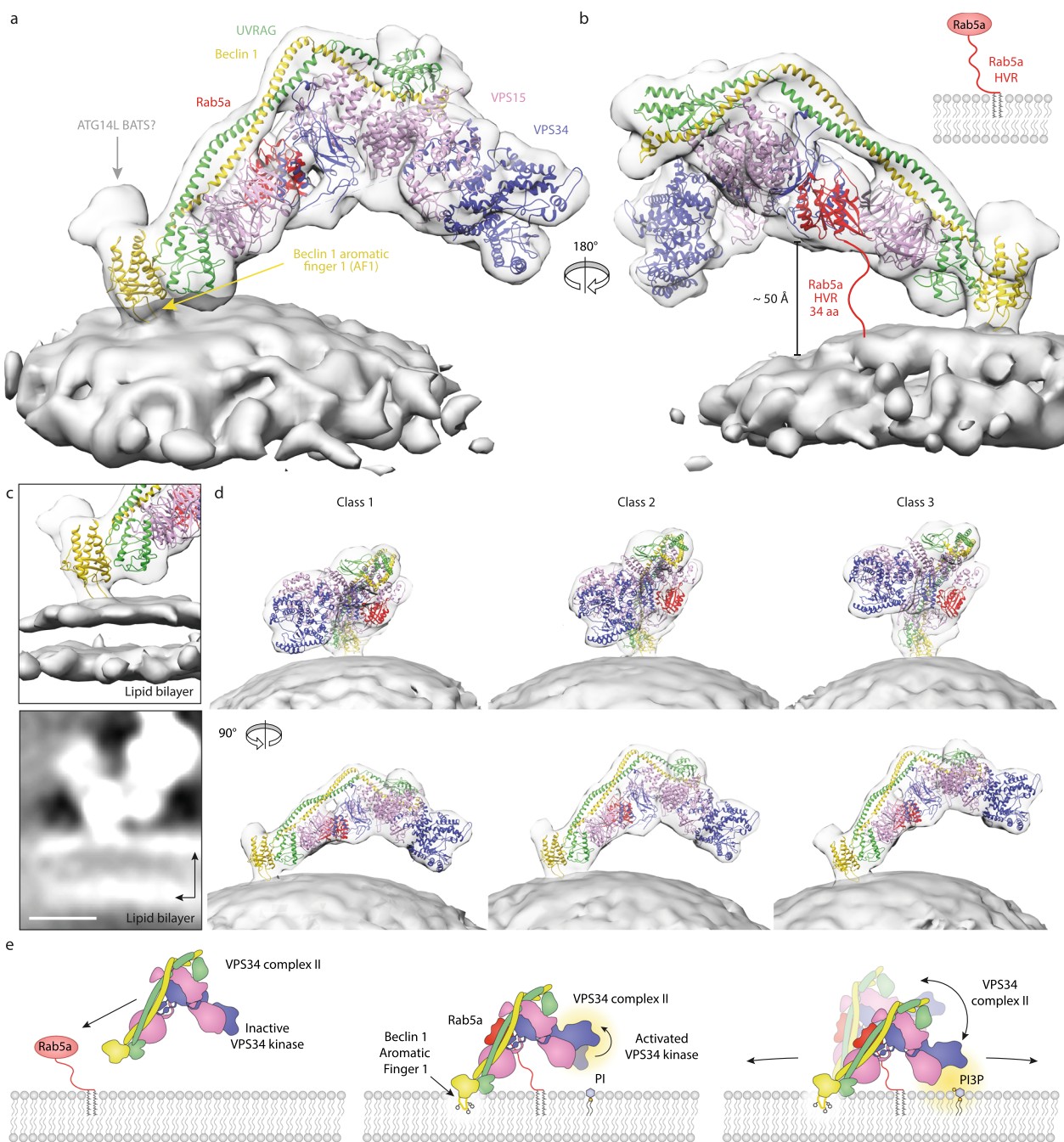

**Fig. 6 The Beclin 1/UVRAG adaptor arm, together with Rab5a, mediates highly flexible membrane binding. a** Subtomogram averaging density of complex II–BATS bound to Rab5a–GTP decorated LUVs. Complex II directly attaches to membranes via its Beclin 1/UVRAG adaptor arm. The catalytic arm hovers above the membrane with no direct contact. No density for the BATS domain is observed. **b** The distance between the Rab5a density and membrane is ~50 Å, which easily can be spanned by the 34 residues forming the Rab5a HVR (not ordered in the structure). **c** The adaptor arm shows density from the Beclin 1 BARA aromatic finger 1 (AF1) in contact with the outer membrane leaflet. Scale bar, 4 nm. **d** Three different classes from 3D classification of Rab5a–GTP/complex II–BATS show different orientations relative to the membrane, however, the same complex II model with two ordered arms and an active conformation of the VPS34 kinase domain can be fit to each class. While the adaptor arm stays bound to the membrane and serves as an anchor point, complex II can tilt both up/down as well as sideways. **e** A three-state model for VPS34 complex II activation on membranes by Rab5 GTPases. Off membranes, the VPS34 kinase domain of complex II is autoinhibited by VPS15 (left). Binding of Rab5a–GTP to the adaptor arm recruits the complex to membranes, releasing the auto-inhibitory interactions and enabling scooting via the adaptor arm state (middle). The complex is able to tilt up and down so that the catalytic arm transiently engages the membrane to phosphorylate PI to synthesise PI3P (right). States 2 and 3 are able to scoot on the membrane and thereby encounter new substrate.

distinguish the contributions of the two complexes to various cellular processes.

Although VPS34 was identified as a critical regulator of cellular trafficking[60] and a key Rab5 effector[13] more than two decades ago, the molecular basis for this remained completely unknown. Our discoveries provide a structural and mechanistic view of how Rab5a recruits and activates complex II on endosomes. Furthermore, we show that Rab1a can do the same for the autophagy-specific complex I, through overlapping but distinct interactions with the core VPS34/VPS15 subunits. This sheds light on the mechanism of Rab1a involvement in autophagy, through a direct activation of complex I. Together, these findings present a framework for future discovery of inhibitors targeting Rab-dependent VPS34 activation for therapeutic purposes. Cryo-ET enabled us to visualise the molecular organisation of Rab5a-GTP/complex II assembly on membranes. The dynamic mode of complex II membrane attachment allows the activated VPS34 kinase to survey the membrane for available substrate and generate local enrichment of PI3P that enables patterning of tightly organised Rab-containing domains[14] that ultimately determine organellar identity.

## Methods

**Plasmids**. Plasmids used in this study are listed in Supplementary Table 2.

**Protein purification of human VPS34, complex I and complex II**. For purifying complexes I and II, Expi293F suspension cells (ThermoFisher A14527) were grown at 37 °C, 8% $CO_2$, and 125 rpm shaking in Expi293 Expression Medium (ThermoFisher A1435102). Plasmids at 1.1 mg/L culture were transfected into the cells at a density of around $2.3 \times 10^6$/mL, using polyethylenimine (PEI) 'MAX' (Polysciences 24765, 1 mg/mL in PBS) at 3 mg/L culture. Cells were grown at the same condition as above for 48 h, then harvested at 3000 g for 25 min, flash-frozen in liquid nitrogen, and stored at −80 °C, until they were used.

To prepare protein, the cells were suspended in 100 mL/2 L cells of lysis buffer (50 mM HEPES, pH 8.0, 150 mM NaCl, 1% Triton X-100 [Sigma, X100], 12% glycerol, 0.5 mM tris[2-carboxyethyl]phosphine (TCEP, Soltec Ventures, M115), 2 mM $MgCl_2$, 1× EDTA-free inhibitor tablet (Roche, 05056489001), and incubated on ice for 30 min. The insoluble fraction was removed by centrifugation at $14,000 \times g$ for 30 min in a Ti45 rotor (Beckman Coulter). The supernatant fraction was incubated with 2 mL of IgG beads (GE Healthcare 17-0969-02) for 3.5 h. The mixture was centrifuged at $1000 \times g$, and 100 mL of supernatant was removed. The IgG beads were transferred to a gravity flow column, and washed with 150 mL of wash buffer (50 mM HEPES, pH 8.0, 150 mM NaCl, 0.1% Triton X-100, 0.5 mM TCEP, 5 mM ATP [Sigma, A2383], 50 mM $MgCl_2$, 5 μg/mL RNaseA [Sigma, 83834]) and 150 mL of TEV buffer (50 mM HEPES, pH 8.0, 150 mM NaCl, 0.5 mM TCEP). A 10 mL aliquot of TEV buffer and 80 μL of 4.4 mg/mL TEV protease were added, and incubated at 4 °C overnight without rotation. The eluate was collected and beads were washed three times with 5 mL buffer each. The elution fractions were combined and concentrated using a 100 kDa concentrator (Millipore, UFC910096) before running on an S200 10/30 column equilibrated with 20 mM HEPES, pH 8.0, 150 mM NaCl, 0.5 mM TCEP. The main peak fractions were pooled and concentrated. The proteins were frozen in liquid nitrogen and stored at −80 °C. Human complex II was purified in the same way as complex I except the NaCl concentration was 300 mM throughout the procedures. All complex I and II mutants were purified in the same way as WT complex I and complex II, respectively.

Purification of human VPS34 was described previously[21]. In brief, a plasmid pSM41 was expressed in bacteria (E. coli C41 (DE3)), purified using Ni-NTA affinity chromatography with sonication buffer (10 mM Tris-HCl pH 8.0, 100 mM NaCl, 10 mM imidazole), 1 Complete EDTA-free Protease Inhibitor Cocktail Tablet (Roche, 11873580001), 0.1 mg/mL DNaseI, and 50 mL BugBuster (Novogen 70584), washed with Ni A1 buffer (20 mM Tris pH 8.0, 300 mM NaCl, 10 mM imidazole, 2 mM β-mercaptoethanol), and Ni A2 buffer (20 mM Tris pH 8.0, 100 mM NaCl, 10 mM imidazole, 2 mM β-mercaptoethanol), and eluted with an imidazole gradient with about 80 mL of Ni B1 buffer (20 mM Tris pH 8.0, 100 mM NaCl, 300 mM imidazole, 2 mM β-mercaptoethanol). The His6 tag was cleaved with TEV protease (made in house) and incubated overnight with gentle rocking at 4 °C, followed by purification on a HiTrap heparin column. The column was washed with 20 mL HA buffer (20 mM HEPES pH 8.0, 100 mM NaCl, 2 mM DTT), and the sample was eluted with 100 mL HB buffer (20 mM HEPES pH 8.0, 2 mM DTT, 1 M NaCl). The protein was further purified by gel filtration in 20 mM HEPES pH 8.0, 100 mM NaCl, 2 mM DTT. The peak fractions were pooled and concentrated to 5.2 mg/mL (51 μM).

**Purification of p40-PX domain**. Purification and labelling of the p40-PX domain were described previously[21]. In brief, a plasmid pYO1125 was expressed in bacteria

(E. coli C41 (DE3)), sonicated in lysis buffer (20 mM HEPES pH 8.0, 200 mM NaCl, 1 mM TCEP, 0.05 μL/mL universal nuclease (ThermoFisher, 88702), 0.5 mg/mL lysozyme (MP Biomedicals, 195303)), affinity-purified with Glutathione Sepharose resin, washed with 100 mL wash buffer (20 mM HEPES pH 8.0, 300 mM NaCl, 1 mM TCEP) and 100 mL TEV buffer (20 mM HEPES pH 8.0, 200 mM NaCl, 1 mM TCEP). The N-terminal GST tag was cleaved with TEV protease and incubated overnight with gentle rocking at 4 °C. The elution fractions were collected and concentrated in a 10000 MWCO Amicon Ultra15 concentrator (Millipore, UFC901024). The concentrated protein was further purified by gel filtration on a Superdex 75 16/60 column (GE Healthcare 17-1068-01), in a buffer containing 20 mM HEPES pH 8.0, 200 mM KCl, 1 mM TCEP. The peak fractions were pooled and concentrated to 23.5 mg/mL (1.35 mM). The purified PX domain was labelled using AF647 C2 Maleimide kit (Life Technologies, A20347), and the labelled protein was purified using a heparin column.

**Purification of human Rab5a**. For the human Rab5a construct that was used for maleimide labelling, a WT Rab5a (1–212) fragment was mutated into Q79L, also surface-exposed cysteines were mutated to serines (C19S-C63S) so that one cysteine was left at the C terminus of the protein (plasmid pOP823 His-SUMO-Rab5a(1–212)-Q79L-C19S-C63S). Protein was overexpressed in E. coli C41(DE3) RIPL purified by Ni-NTA FF columns (GE Healthcare 17-5255-01), followed by the removal of the His-tag with SUMO protease, dialysis overnight (10 kDa MWCO, SnakeSkin™ ThermoFisher) and another passage through Ni-NTA resin. The flow-through was concentrated and mixed with 11 molar excess of GTP (Jena bioscience NU-1012) or GDP (Jena bioscience NU-1172) and 18 molar excess of EDTA for 90 min at room temperature. The $MgCl_2$ was then added to 36 molar excess and incubated for another 30 min. The mixture was loaded on a size-exclusion chromatography and the peak fractions were concentrated to ~0.5–1 mM in 25 mM HEPES pH 7.0, 150 mM NaCl, 0.5 mM tris-(2-carboxyethyl) phosphine (TCEP).

For HXD-MS, a Rab5a-Q79L construct was designed with last 4 residues (CCSN) deleted (plasmid pYO1261 His-SUMO-Rab5a(1-211)-Q79L) and was purified as above.

**Purification of human Rab1a**. For the human Rab1a construct (plasmid pJB78 GST-TEV-Rab1a(1–204)-Q70L-C26S-C126S) that was used for maleimide labelling, a WT Rab1a (1–204) fragment was mutated into Q70L also the surface-exposed cysteines were mutated to serines (C26S-C126S) so that only one Cys was left at the C terminus of the protein. Protein was overexpressed in E. coli C41(DE3) RIPL, purified on Glutathione Sepharose 4B resin (GE Healthcare 17-0756-05), followed by the removal of the GST tag with TEV protease overnight. The cleaved protein was diluted to a final concentration of 50–100 mM NaCl in dilution buffer (20 mM HEPES 8.0 and 1 mM TCEP) and passed through a 5 mL HiTrap Q column (GE Healthcare, 17505301) followed by a 5 mL HiTrap Heparin column (GE Healthcare, 17040601) to remove cleaved GST and TEV protease. The heparin flow-through was concentrated, loaded with nucleotide and $MgCl_2$, purified and concentrated in the same way as Rab5a.

For HXD-MS, a Rab1a–Q70L construct was designed with last 2 residues (CC) deleted. This construct (plasmid pYO1262 His-TEV-Rab1a(1-203)-Q70L) was expressed and purified by affinity chromatography on Ni-NTA, ion-exchange chromatography on Q column and Heparin column and gel filtration on Superdex 75 16/60.

**Synthesis of BrCO6K**. We synthesised BrCO6K for incorporation into Rab5a using the scheme illustrated in Supplementary Fig. 8. 4-Bromobutan-1-ol (1.0 g, 6.5 mmol, 1 eq.) was dissolved in anhydrous dichloromethane (DCM) (13 mL) and cooled to 0 °C in an ice bath. N,N'-disuccinimidyl carbonate (2.0 g, 7.9 mmol, 1.2 eq.) and DIPEA (2.3 mL, 13.0 mmol, 2 eq.) were added at 0 °C. The reaction was allowed to warm up to room temperature and was stirred for 4 h until thin layer chromatography (TLC) showed full conversion of the starting material. Then anhydrous tetrahydrofuran (THF) (13 mL) and α-Boc-Lys-OH (1.9 g, 7.9 mmol, 1.2 eq.) were added and the reaction was stirred overnight at rt. The reaction mixture was diluted with DCM and the organic phase was washed twice with 10% citric acid and once with brine and the combined organic phase was dried over $Na_2SO_4$. The solvent was removed under reduced pressure and the crude product was purified by flash column chromatography (DCM:MeOH = 97:3 + 0.25% acidic acid). 1.6 g (57% yield) α-BocBrCO6K were isolated as light yellow oil.

$^1$H NMR (500 MHz, DMSO-d$_6$) $\delta$ = 7.07 (t, $^3J$ = 5.7 Hz, 1H), 7.00 (d, $^3J$ = 7.9 Hz, 1H), 3.95 (t, $^3J$ = 6.5 Hz, 2H), 3.88–3.77 (m, 1H), 3.55 (t, $^3J$ = 6.6 Hz, 2H), 2.93 (q, $^3J$ = 6.6 Hz, 2H), 1.84 (p, $^3J$ = 6.8 Hz, 2H), 1.70–1.49 (m, 4H), 1.42–1.23 (m, 13H) ppm.

$^{13}$C NMR (75 MHz, MeOD) $\delta$ = 176.2, 159.1, 158.2, 80.5, 64.8, 54.8, 41.4, 33.9, 32.4, 30.6, 30.5, 28.9, 28.7, 24.1 ppm.

MS(ESI), m/z calced. for $C_{16}H_{29}BrN_2O_6$: 424.12 (for $^{79}Br$), 426.12 (for $^{79}Br$); found: 447.1 [M+Na]$^+$ (for $^{79}Br$), 449.1 [M+Na]$^+$ (for $^{81}Br$).

α-Boc-BrCO6K (1.6 g, 3.7 mmol, 1 eq.) was dissolved in DCM (40 mL) and TFA (40 mL) was added. The reaction was stirred overnight at room temperature and the solvent was removed under reduced pressure. The crude product was dissolved in methanol (2 mL) and the solution added dropwise to cold diethylether

(−20 °C, 80 mL). The suspension was centrifuged at 3234 g at 4 °C, the supernatant was discarded and the solid was collected, resuspended and again washed with cold diethylether, twice. The product was dried on air and 0.87 g (72% yield) BrCO6K were isolated as a light brown solid.

**Site-specific incorporation of BrCO6K into Rab5a in bacteria.** BrCO6K was synthesised as described above. In contrast to previously reported BrC6K[24], BrCO6K bears a carbamate moiety linking the ε-amino group of lysine to the bromoalkyl functionality, making it stable against the *E. coli* deacylase CobB. The expression of Rab5a (plasmid STp6, pBAD_Rab5a_Q79L_1-212_C19S_C63S_S84TAG-His6) with site-specifically incorporated BrCO6K was carried out as described previously[24]. Protein was overexpressed in DH10B cells (Invitrogen, 18290015) and purified by Ni-NTA FF columns (GE Healthcare 17-5255-01). The protein was concentrated, loaded with nucleotide/MgCl₂ as described above and further purified size-exclusion chromatography. The peak fractions were concentrated to ~0.5–1 mM in 25 mM HEPES pH 8.0, 150 mM NaCl, 0.5 mM tris-(2-carboxyethyl) phosphine (TCEP).

**Crosslinking of Rab5a S84BrCO6K to VPS34 and complex II.** Purified complex II was mixed with 50–100 molar excess of purified Rab5a S84BrCO6K in 5–15 μL reaction volume. VPS34 was mixed with 100–200 molar excess of Rab5a S84BrCO6K in 5–15 μL reaction volume. Both samples were incubated overnight at 4 °C and then analysed by SDS-PAGE. The gel bands of crosslinked products were cut out and analysed by mass spectrometry.

The excised gel bands were digested as described previously with minor modifications[61]. Trypsin was prepared in the digestion buffer (45 mM ammonium bicarbonate, 10% acetonitrile, v/v) at a concentration of 5 ng/μL. The digestion was incubated at 37 °C for 15 h. Resulting peptides were de-salted using C18 StageTips[62].

Liquid chromatography–tandem mass spectrometry (LC–MS/MS) analysis was performed using an Orbitrap Fusion Lumos Tribrid mass spectrometer (ThermoFisher Scientific), connected to an Ultimate 3000 RSLCnano system (Dionex, ThermoFisher Scientific). Peptides were injected onto a 50-cm EASY-Spray C18 LC column (ThermoScientific) that is operated at 50 °C column temperature. Mobile phase A consists of water, 0.1% v/v formic acid and mobile phase B consists of 80% v/v acetonitrile and 0.1% v/v formic acid. Peptides were separated using a linear gradient going from 2% mobile phase B to 40% mobile phase B over 110 min, followed by a linear increase from 40 to 95% mobile phase B in 11 min. Peptides were loaded and separated at a flow rate of 0.3 μL/min. Eluted peptides were ionised by an EASY-Spray source (ThermoScientific) and introduced directly into the mass spectrometer. The MS data is acquired in the data-dependent mode with a 3-s acquisition cycle. The full scan mass spectrum was recorded in the Orbitrap with a resolution of 120,000. The ions with a charge state from 3+ to 7+ were isolated and fragmented using higher-energy collisional dissociation (HCD). For each isolated precursor, one of three collision energy settings (26%, 28% or 30%) was selected for fragmentation using data-dependent decision tree based on the *m/z* and charge of the precursor. The fragmentation spectra were then recorded in the Orbitrap with a resolution of 50,000. Dynamic exclusion was enabled with a single repeat count and 60-s exclusion duration.

The MS2 peak lists were generated from the raw mass spectrometric data files using the MSConvert module in ProteoWizard (version 3.0.11729). The default parameters were applied, except that Top MS/MS Peaks per 100 Da was set to 20 and the de-noising function was enabled. Precursor and fragment *m/z* values were recalibrated. Identification of cross-linked peptides was carried out using xiSEARCH software (https://www.rappsilberlab.org/software/xisearch)[63]. For crosslinked Rab5a S84BrCO6K-VPS34 sample, the peak list was searched against the sequences and the reversed sequences of Rab5a and VPS34, and for crosslinked Rab5a S84BrCO6K-complex II sample, the peak list was searched against the sequences and the reversed sequences of Rab5a and all subunits of complex II. For both cases, BrCO6K (residue 84) was defined as a modified lysine residue (Kbrco6k) in the Rab5a sequence file. The crosslinking specificity was defined between BrCO6K and any cysteine, lysine, histidine, aspartate and glutamate residues. The following parameters were applied for the search: MS accuracy = 4 ppm; MS2 accuracy = 10 ppm; enzyme = trypsin (with full tryptic specificity); allowed number of missed cleavages = 4; missing monoisotopic peak = 2; fixed modifications = carbamidomethylation on cysteine; variable modifications = oxidation on methionine. All crosslink-spectrum matches (CSMs) that were autovalidated by xiSEARCH were further manually inspected and validated. No CSM against the reversed protein sequences (decoy matches) were autovalidated. A list of identified cross-linked peptide pairs is reported in Supplementary Table 1 and an annotated fragmentation spectrum can be seen in (Supplementary Fig. 9).

**Hydrogen/deuterium exchange mass spectrometry (HDX-MS).** Complex II (5 μM) alone and a mixture of complex II (5 μM) with 30 μM Rab5a-Q79L-GTP were incubated for ~30 min at room temperature. An aliquot of 5 μL of complex II alone was exposed to 45 μL of D₂O Buffer only (25 mM HEPES pH 8.0, 50 mM NaCl, 1 mM TCEP and D₂O at 94.2% final concentration (D₂O, Acros Organics 351430075)) for a defined period of time at room temperature. Furthermore, an aliquot of 5 μL complex II (5 μM) with 30 μM Rab5a-Q79L-GTP exposed to 45 μL

D₂O Buffer + Rab5 (25 mM HEPES pH 8.0, 50 mM NaCl, 1 mM TCEP, 30 μM Rab5a–GTP and D₂O at 94.2% final concentration) for a defined period of time at room temperature. The final D₂O sample concentration was 84.8% D₂O. Four time-points were produced (3/30/300/3000 s), with each exchange reaction executed in triplicate. The exchange reaction was quenched using 20 μL of ice-cold 5 M guanidinium chloride, 15 mM TCEP, and 8.4% formic acid, pH 1 in UPLC grade H₂O (Romil, H949). The final pH of the sample was 2.2. Each sample was immediately flash-frozen in liquid nitrogen and subsequently stored at −80 °C until analysis.

Complex I (7 μM) alone and a mixture of complex I (7 μM) with 28 μM Rab1a–Q70L–GTP were incubated for 10 min at room temperature. An aliquot of 5 μL of this stock solution was mixed with 40 μL of D₂O buffer (consisting of 20 mM HEPES pH 8.0, 150 mM NaCl, 0.5 mM TCEP in D₂O, final concentration 94.7%) for a defined period of time at room temperature. The following procedures are essentially the same as complex II + Rab5a.

Samples were quickly thawed and manually injected on an M-Class Acquity UPLC with HDX Manager technology (Waters) set to maintain a constant temperature of 0.1 °C. Proteins were digested using an in-line Enzymate Immobilised Pepsin Column (Waters) at 15 °C for 2 min, and were collected on a van-guard pre-column trap (Waters). Digested peptides were eluted from the trap onto an Acquity 1.7 μm particle, 100 mm × 1 mm C18 UPLC column (Waters), equilibrated in Pepsin-A buffer (0.1% formic acid), using a 5–36% gradient of Pepsin-B buffer (0.1% formic acid, 99.9% acetonitrile) over 26 min. Peptide data were collected using a Waters Synapt G2 Si (Waters) over a 50–2000 *m/z* range using the High-Definition MSe data acquisition mode fitted with an ESI source.

Peptide identification was done with ProteinLynx Global Server (PLGS, Waters, UK). Peptides were identified from four non-deuterated samples for complex I, and three non-deuterated samples for complex II. Deuterated peptides were analysed by DynamX 3.0 software (Waters, UK). Peptide inclusion criteria was a minimum score of 6.4, a minimum of 0.3 products per amino acid, a maximum MH+ error of 5 ppm, and a positive identification meeting these criteria in at least 2 of the 3 non-deuterated files. An initial automated spectral processing step was conducted by DynamX followed by a manual inspection of individual peptides for sufficient quality. A table of every peptide included within the dataset and the quality assessment statistics of the dataset are available in Supplementary Data 1 and 2. The HDX-MS analysis in this manuscript complies with the community agreed guidelines[64].

**GUV assays and Rab labelling of GUVs.** A lipid mixture was assembled according to Supplementary Table 3. The procedures for GUV generation and immobilisation on an observation chamber were described previously[21]. For GUV generation, a 15 μL aliquot of a 1 mg/mL GUV lipid mixture in chloroform was placed onto the Indium-Tin-Oxide (ITO)-coated side of an ITO slide (Nanion) then dried in a desiccator for 1 h. GUVs were made in the presence of 220 μL of swelling solution (0.5 M sucrose), using a GUV maker (Vesicle Pro, Nanion) with a programme at 10 Hz; 60 °C; 1 Amp; 3 min rise; 68 min fall. After GUVs were produced, they were immediately removed from the slide and transferred to a 1.5 mL tube, which had been coated with 5 mg/mL BSA (Sigma A7030) for 1 h then rinsed once with a swelling solution. For GUV immobilisation, wells of an eight well glass bottom chamber (Ibidi 80827) were coated with 100 μL of avidin solution (0.1 mg/mL avidin egg white, Life Technologies A2667 dissolved in PBS, and 1 mg/mL BSA) for 15 min, then washed two times with observation buffer (25 mM HEPES pH 8.0 and 271.4 mM NaCl). An aliquot of observation buffer was added to the wells, followed by the addition of the GUVs. Rabs (with a C-terminal Cys) were incorporated onto GUVs by adding 10 μM of Rab (fourfold molar excess over MCC-PE lipid) to the immobilised GUVs and incubating at 4 °C overnight. Unbound Rabs were removed by carefully adding and taking off 360 μL wash buffer five times (31.8 mM HEPES pH 8.0, 172.7 mM NaCl, 5 mM β-mercaptoethanol, and 181.8 mM sucrose). The kinase reaction, microscopy, and image analysis were described previously[21]. GUVs were observed with a ×63 oil immersion objective (Plan-Apochromat ×63/1.40 Oil DIC, Zeiss) on an inverted confocal microscope (Zeiss 780), using ZEN software (Zeiss). The observation chamber was immobilised on a microscope stage holder using an adhesive (Blu-Tack, Bostik). In the ZEN software, Time Series and Positions were selected. The Lissamine-rhodamine channel for GUVs was excited with a Diode-pumped solid-state (DPSS) 561 nm laser and collected with a 566–629 nm band. The Alexa Fluor 488 channel for the secondary antibody was excited using an Argon multiple 458, 488 and 514 nm laser, and collected with a 500–530 nm band. Reaction progress curves for the kinase assays with SDs for each time point are shown in Supplementary Fig. 10.

For Rab5 immunostaining on the GUVs, rabbit Anti-Rab5 Antibody (#2143S Cell Signalling) was added after Rab labelling at 1:100 dilution and left overnight at 4 °C. Unbound primary antibodies were removed by adding and taking off 360 μL wash buffer four times. Goat anti-rabbit secondary antibody Alexa Fluor 488 (ThermoFisher Catalogue #: A-11008) was added at 1:250 dilution and incubated for 2 h at room temperature. Unbound secondary antibodies were removed by adding and taking off 360 μL wash buffer four times.

**LUV preparation.** A lipid mixture was assembled according to Supplementary Table 3 and dried under nitrogen gas. The glass vial was rotated so that a thin film

was formed on the glass wall. The remaining solvent was evaporated under vacuum for 1 h. The lipids were dissolved in lipid buffer (25 mM HEPES pH 8.0, 150 mM NaCl, 1 mM TCEP or for maleimide reactions in 25 mM Hepes pH 7.0, 150 mM NaCl) and vortexed for 2 min. The solution was transferred to a 1.5 mL Eppendorf tube and sonicated for 2 min in a bath sonicator. After 10 cycles of freeze/thaw in liquid nitrogen and a 43 °C water bath, the lipid mixture was extruded at least 20 times through a 100 nm filter for flotation assays (Whatman Anotop 10 syringe filter 0.1 μm pore size, 10 mm diameter, Cat No 6809-1112) or 50 nm filter for cryo-ET (NanoSizer MINI Liposome Extruder, Part Code: TT-001-0010). The lipid solution was then used fresh.

**Rab immobilisation on LUVs.** The LUVs included Lissamine Rhodamine (0.2%) in order to measure the LUV concentration throughout the LUV preparation by measuring the emission (ex 560/em 583) on a plate reader (PHERAstar BMG LABTECH). To ~3 mM LUVs (containing 0.2 mM or 6% PE-MCC), Rab was added in 0.3–0.6× molar ratio to PE-MCC lipid in 25 mM Hepes pH 7.0, 150 mM NaCl. After overnight incubation in the fridge, the reaction was spun for 5 min at $5000 \times g$ to remove any precipitate. Then, LUVs were pelleted by centrifuging the supernatant for 30 min at $60,000 \times g$ at 4 °C in a TLA 100 rotor (Beckman Coulter). After the spin, the supernatant, which has unreacted Rab, was taken off and the clearly visible pellet containing LUVs was re-dissolved in the buffer (25 mM Hepes pH 8.0, 150 mM NaCl, 1 mM TCEP) and the final LUV concentration was determined by Lissamine Rhodamine emission.

**Flotation assay.** The mixture of proteins and lipids contained 1.8 mM (~1.5 mg/ mL) LUVs and 2 μM VPS34 complex in buffer containing 25 mM HEPES pH 8.0, 150 mM NaCl and 1 mM TCEP. The total sample volume was 20 μL. While the LUVs and proteins were incubated on ice for 30 min, a sucrose gradient was prepared. For the gradient, several sucrose solutions were layered from the bottom to the top in a Beckman centrifuge tube (343775 Thickwall Polycarbonate Tube, Beckman Coulter): 40 μL 30% sucrose solution, 52 μL 25% sucrose solution, 52 μL 20% sucrose solution. Then 16 μL of the LUV/protein sample was carefully pipetted on top of the gradient. From the remaining LUV/protein sample, 2.5 μL were kept as an input sample for the SDS-PAGE. The gradient was then centrifuged for 3 h in a TLS-55 rotor (Beckman Coulter) at $258,500 \times g$ at 4 °C. Afterwards, 6 fractions of 26 μL each were carefully collected from the top of the gradient. The input and gradient fractions were analysed by SDS-PAGE.

**MitoID.** The MitoID experiments were performed following the methods previously described[28]. Briefly, two confluent T175 flasks of HEK293T cells were transfected with the Rab-BirA*-HA-MAO chimeras for 24 h, after which exogenous biotin was added for another 16 h. Cells were then washed and lysed, using 25 mM Tris pH 7.4, 150 mM NaCl, 1 mM EDTA, 1% (v/v) Triton X-100, 1 mM PMSF, and 1 cOmplete protease inhibitor cocktail tablet (Roche) per 50 mL buffer. After 30 min of incubation, the lysate was added to washed MyOne Streptavidin Dynabeads (Invitrogen) and left to rotate at 4 °C overnight. After ~16 h, beads were repeatedly washed in Wash Buffer 1 (2% SDS and cOmplete protease inhibitors), Wash Buffer 2 (1% (v/v) Triton X-100, 0.1% (w/v) deoxycholate, 500 mM NaCl, 1 mM EDTA, 50 mM HEPES or Tris pH 7.4 and cOmplete protease inhibitors), and Wash Buffer 3 (50 mM Tris pH 7.4, 50 mM NaCl, and cOmplete protease inhibitors). Finally, the beads were eluted in SDS sample buffer and 3 mM biotin by incubating for 3 min at 98 °C, after which 1 mM β-mercaptoethanol was added. Samples were analysed by tandem LC–MS/MS mass spectrometry, as described[28], as well as loaded onto a Tris-Glycine gel and transferred onto nitrocellulose paper. Western blots were blocked in 5% (w/v) milk and probed with primary and secondary antibodies, all for 1 h each and with 0.01% PBS-Tween-20 washes in between. The antibodies used were mouse anti-HA (supernatant from hybridoma clone 12CA5 used at a dilution of 1:250), rabbit anti-VPS15 (Proteintech, 17894-1-AP, dilution 1:1000), rabbit anti-VPS34 (Protein-tech, 12452-1-AP, dilution 1:1000), rabbit anti-Beclin 1 (Santa Cruz Biotechnology, sc-11427, lot #H3012, dilution 1:1000), rabbit anti-ATG14L (Cell Signaling Technology, 5504S, dilution 1:750), and rabbit anti-UVRAG (Cell Signaling Technology, 5320S, dilution 1:1000). Uncropped western blots can be seen in Source Data for Fig. 3a.

**Cell culture and confocal microscopy.** HEK293T cells were grown in DMEM medium (Life Technologies, 31966047) containing 10% FBS (ThermoFisher, 10270106) and 1% Pen–Strep (Gibco, 15140122) at 37 °C with 5% CO₂ until 60–70% confluency. After stripping the cells using 0.05% trypsin in 0.5 mM EDTA, the cell density was measured by a cell counter (ThermoFisher, AMQAX1000), then adjusted to $0.3 \times 10^6$/mL in the same medium. Diluted cells (0.5 mL) were grown on a coverslip in a well of a 24-well plate (Corning, 3526) for 24 h at 37 °C with 5% CO₂. The medium was replaced with pre-warmed Opti-MEM medium (ThermoFisher, 11058021) containing 10% FBS, and grown for 6 h at 37 °C with 5% CO₂ for DNA transfection.

Cells were transfected with 1.1 μg/mL plasmid DNA and 3.3 μg/mL polyethylenimine (PEI) 'MAX' (Polysciences 24765, 1 mg/mL in PBS). The concentration of each plasmid for multiple subunits was initially adjusted to 100 ng/μL in PBS, then equal amount of DNA was mixed to make up to 1.1 μg/mL.

The DNA and PEI were mixed in 20 μL PBS (total volume) for 5 min at RT, then added to the cells, and incubated for 18–24 h at 37 °C with 5% CO₂.

Cell were fixed with 4% paraformaldehyde at RT for 10 min, washed once with PBS, permeabilised with 0.05% digitonin at RT for 10 min, then washed three times with PBS. The coverslips were located upside down on a slide glass which had been spotted with a mounting medium (Vector Laboratories, H-1000). The coverslips were immobilised by nail varnish. Cells were observed using an inverted confocal microscope (Zeiss 780) with a ×63 oil immersion objective (Plan-Apochromat ×63/ 1.40 Oil DIC, Zeiss) using ZEN software (Zeiss). The mCherry channel was exited with a Diode-pumped solid-state (DPSS) 561 nm laser and collected with a 600–630 nm band. The EGFP channel was excited with an Argon multiple 458, 488 and 514 nm laser, and collected with a 500–530 nm band. Micrographs were analysed by Fiji and Adobe Photoshop (Adobe).

For the quantification of VPS34-GFP localisation at the mCherry-Rab peak position in the transects in Figs. 2e and 4e, the value in the GFP channel at the mCherry-Rab peak was divided by the mean background in the GFP channel for 3–10 regions outside the peak area. More than three fields of cells and $n = 15$ cells were included for each analysis. The image analysis was done using Fiji (ImageJ), Microsoft Excel, and GraphPad Prism7.

**Cryo-ET and subtomogram averaging.** LUVs were labelled with Rab5a–GTP as described above. For grid preparation, LUVs were mixed with complex II–BATS and incubated on ice for 30–45 min. BSA-coated gold fiducials (Gold nanoparticles 10 nm, BBI Solutions EM.GC10) were then added up to the final volume of 3 μL and mixed (6 mg/mL LUVs and 16 μM complex II–BATS, final concentrations). The 3 μL samples were applied immediately to multi-hole grids (Multi A (various hole sizes, carbon film, Grid: Au, Mesh: 300, QUANTIFOIL). The grids were glow discharged for 30 s with the Quorum SC7620 glow discharger prior to use. The samples were plunge-frozen in liquid ethane cooled by liquid nitrogen inside a Vitrobot (FEI, ThermoFisher). Blotting papers were left at least for 30 min in 100% humidity at 18 °C. The Vitrobot blot force was 20, with a blot time of 6 s.

Tomogram acquisition for complex II–BATS on Rab5a–GTP decorated membranes was performed on an FEI Titan Krios electron microscope (ThermoFisher Scientific) operated at 300 kV with a Gatan Quantum energy filter (slit width 20 eV) and a K3 direct detector operated in counting mode. Serial-EM software was used to acquire each tilt series using a dose-symmetric scheme with a tilt range ± 60°, 3° angular increment and defoci from −2 to −5 μm[65,66]. Each tilt series of 41 10-frame movies was recorded in counting mode with a pixel size of 2.133 Å/px at a dose rate of ~5.5 e⁻/Å²/s and a total dose per tomogram of ~123 e⁻/Å². A total of 115 tilt series were collected over 2 days. Data collection parameters are summarised in Supplementary Table 4.

The raw movies were corrected for detector gain pixel defects and aligned using 'alignframes' from the IMOD package[67] (installed using Python 3.7). Tilt series that could not be aligned during gold fiducial alignment or had contamination such as dirt or ice were discarded. Tilt-series were low pass filtered according to the cumulative radiation dose[68] and aligned using gold fiducial markers in the IMOD package. Bin8 (pixel size 17.064 Å/px) and bin4 (pixel size 8.532 Å/px) non-contrast-transfer-function (CTF) corrected tomograms were reconstructed by weighted back-projection in IMOD. 3D CTF-correction for bin2 (pixel size 4.266 Å/px) and bin1 (pixel size 2.133 Å/px) tomograms were performed using NovaCTF phaseflip[69] with defocus estimation by 'ctfplotter' from the IMOD package. EMAN2.2 was used for converting map formats.

Subtomogram alignment and averaging were done as previously described[37,70] using the subTOM package written with MATLAB (MathWorks) functions adapted from the TOM[71], AV3[72] and Dynamo packages[73]. The scripts and relevant documentation are available to download [https://www2.mrc-lmb.cam.ac.uk/ groups/briggs/resources] and [https://github.com/DustinMorado/subTOM/releases/ tag/v1.1.4]. Additionally, instead of a binary wedge mask, a modified wedge mask was used[70]. The missing wedge was modelled at all processing stages as the sum of the amplitude spectra of subtomograms extracted from regions of each tomogram containing empty ice, and was applied during alignment and averaging.

To define the initial subtomogram positions, a plugin for Chimera was used to set centres and radii of vesicles, which had a dense protein coat, in 105 bin8 tomograms[74]. A wide range of sizes of vesicles was used in the analyses, and only vesicles with invaginations or large deformities were omitted. The subtomogram coordinates (x/y/z) and initial Euler angles (but random in-plane rotation) were defined by the sphere surface with a uniform sampling of 8 px (~136 Å). As vesicles are not perfectly spherical, subtomograms were first aligned to the vesicle lipid bilayer with a reference containing a membrane bilayer. The vesicle diameters were determined by calculating the vesicle centroid by the average position of the subtomograms around each vesicle after membrane alignment. Then an average diameter was calculated by measuring the distance of each subtomogram to the calculated centroid and multiplied by two. The distance between the protein complex and membrane (~6 nm) was subtracted from the calculated diameter. To identify particles, subtomograms were aligned to an initial reference consisting of two gaussian-filtered ellipsoids forming a V-shape (Supplementary Fig. 6a, reference). After aligning against the V-shape, some subtomograms converged and formed clusters, which indicated the presence of a particle (Supplementary Fig. 6a, A). The subtomogram coordinates were cleaned by a minimal distance threshold (distance cutoff 8 px, cluster size 2, cluster distance 2 px) and cross-correlation

cutoff so that 191,169 particles remained (Supplementary Fig. 6a, B). Subtomograms extracted from the very top and bottom of the vesicles, where they contact the air-water interface, were excluded by the cross-correlation cutoff. Subtomograms were split into even/odd halves in bin4 and further aligned separately until the resolution did not improve (Supplementary Fig. 6a, C). In order to analyse the heterogeneity of the subtomograms, a principal component analysis (PCA) classification on wedge-masked difference maps was used to classify the protein complex region within the subtomograms[52]. For the PCA, a cylindrical mask was applied that encompassed the protein density and excluded the membrane density. The first 5 eigencomponents were used to sort the data into 20 classes (Supplementary Fig. 6b). Of the 20 classes, classes 1–6 showed the most distinct features as well as the presence of the membrane and were combined to a total of 26,979 particles and further aligned in bin4 (Supplementary Fig. 6a, D). The alignment was then continued in bin2, where the subtomograms were shifted to the centre of the box, and bin1 until no improvement in resolution could be achieved (Supplementary Fig. 6a, E and F). The local resolution calculated in Relion 3.0[75] showed a range of 8–16 Å with an overall resolution of 9.8 Å (FSC 0.143 cutoff) (Supplementary Fig. 5). Additionally, the LAFTER algorithm was used for local de-noising of the final maps[76]. In order to classify the complex according to its orientation relative to the membrane, another PCA classification was performed on bin4 subtomograms, applying a cylindrical alignment mask that encompassed the membrane density and excluded the protein density. Classes 1–3 were subsequently further aligned in bin2.

**Model building**. As no map coordinates have been deposited for the cryo-EM structures by Chang et al. or Young et al., we have used the SWISSMODEL homology-modelling server[41] and the crystal structure of yeast complex II (PDB: 5DFZ) to generate an initial model. UCSF Chimera[77] was used for rigid-body fitting and visualisation. Following this, the kinase domains of the complex II subunits were adjusted with Coot as rigid bodies to fit the EM density[78]. An initial model for the human VPS34 HELCAT was built starting with PDB ID 3IHY. PDB entry 4DDP was used as an initial model for the human Beclin 1 BARA domain[50]. PDB ID 3MJH provided an initial model for the Rab5a[47]. Some manual adjustments were made to the density using Coot[45,78,79], then the structure was regularised and fit to the density using REFMAC[42–45]. Finally, the model of complex II was refined using molecular dynamics in ISOLDE/ChimeraX[46,80], while manually adjusting the model to improve the geometry and fit to the density. The model was subsequently refined with REFMAC. Prior to our work reported here, the only structure that was reported for a human VPS34 complex with both arms of the V-shaped complex ordered was the cryo-EM reconstruction of the NRBF2 MIT domain in a complex with VPS34 complex I[34]. Although It was reported that the yeast complex II could be fit into the density for this complex I reconstruction, no coordinates were deposited. Consequently, we fit a model for complex I into the complex I cryo-EM density (EMD-20390). This required some manual readjustment of the model, using the programme COOT[78,79] followed by REFMAC refinement with PROSMART[42] restraints to improve the geometry. This model of complex I enabled us to better interpret the HDX-MS observations that suggest how Rab1a interacts with complex I.

**Reporting summary**. Further information on research design is available in the Nature Research Reporting Summary linked to this article.

## Data availability

The cryo-ET structure and a representative tomogram are deposited in the Electron Microscopy Data Bank (EMDB) under accession codes EMD-12214 (3D cryo-ET map with the membrane masked-out before B-factor sharpening in LAFTER), EMD-12237 (a reconstruction in which the lipid membrane was not masked-out before sharpening in LAFTER) and EMD-12238 (a sample non-CTF-corrected bin4 tomogram). The associated molecular model is deposited in the protein data bank (PDB) under accession codes 7BL1. Protein structures from published work that were used in this study are available in the PDB under accession codes: 5DFZ, 3IHY, 4DDP and 3MJH. CLMS data were deposited to ProteomeXchange (aka PRIDE) (accession code PXD023533) and jPOST (accession code JPST001056). The MATLAB package subTOM is available for download from https://www2.mrc-lmb.cam.ac.uk/groups/briggs/resources and https://github.com/DustinMorado/subTOM/releases/tag/v1.1.4. All reagents generated by this study are available from the corresponding authors on request. Source data are provided with this paper.

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

## Acknowledgements

We thank Matthias Stahl, Paul Emsley, Garib Murshudov, the MRC-LMB EM facility, Jake Grimmett, Toby Darling, Dina Schneidman and Merav Braitbard for assistance and advice, Melissa Gammons and Maki Ohashi for advice on cell biology experiments and Glenn Masson for advice with HDX-MS. The work was supported by the Medical Research Council (MC_U105184308 to RLW, MC_U10517878 to S.M. and MC_UP_1201/16 to J.A.G.B.), Cancer Research UK (grant C14801/A21211 to R.L.W.), the European Research Council (ERC) under the European Union's Horizon 2020 research and innovation pro-gramme (ERC-CoG-648432 MEMBRANEFUSION to J.A.G.B.), the framework of SFB1035 (German Research Foundation DFG, Sonderforschungsbereich 1035, project number 201302640, project B10 to K.L.), the Deutsche Forschungsgemeinschaft (DFG, German Research Foundation) (No. 392923329 to J.R.) and the Wellcome Trust through a Senior Research Fellowship (No. 103139 to J.R.). The Wellcome Centre for Cell Biology is sup-ported by core funding from the Wellcome Trust (No. 203149 to J.R.).

## Author contributions

S.T., Y.O., D.M., J.B., O.P., L.T.L.B., M.-K.v.W., S.L.M. and O.K. conducted the research. S.T., Y.O., D.M., J.B., O.P., O.K., K.L., S.M., J.AG.B. and R.L.W. analysed data. S.T., Y.O., D.M., O.P., K.L., S.M., J.A.G.B. and R.L.W. developed the experimental plan. J.R. and Z.A.C. carried out crosslinking/mass spectrometry analysis. S.T., Y.O., O.P. and R.L.W. wrote the draft. All authors reviewed and edited the drafts.

## Competing interests

The authors declare no competing interests.
