## [Peer Review File · Nature Communications]

REVIEWER COMMENTS

Reviewer #1 (Remarks to the Author):

The manuscript by Tremel et al. investigates the structural basis of VPS34 activation on membranes. The authors established that Rab5a is a potent activator of the heterotetrameric core complex II containing VPS34 and mapped the Rab5a-GTP binding site on complex II using crosslinking and hydrogen deuterium exchange mass spectrometry. Also, expanding MitoID analysis to include Rab1 the authors detected all four subunit of the VPS34 Complex 1 and further demonstrated that Rab1 interacts and activates VPS34 complex I. Furthermore, the authors did Cryo-ET to establish the structure of VPS34 Complex II on RAB5-GTP positive vesicles and elucidated the key steps of the activation of VPS34 complex II by Rab5.

The manuscript is clearly written and presents a number of novel and interesting data mainly on the interaction between Rab5 and VPS34 complex II and its activation. Experiments are carefully performed with the appropriate controls and conclusions are based on solid and convincing results.

Thus, I believe that this manuscript deserves publication in Nature Communications although I have one general criticism regarding the part of the manuscript dealing with Rab1 that should be addressed before publication. While the authors go very deep in the definition of the steps regarding recruitment, binding and activation of VPS34 Complex II by Rab5, the interesting results obtained on Rab1 and VPS34 Complex I are somehow not pursued. Indeed, the results show clearly that Rab1 interacts and activates the autophagy associated VPS34 complex I, but no experiments are shown regarding the structural basis of this interaction and activation. It is clear that the authors cannot do all the experiments done on Rab5 during the short time of a revision but they could try to perform just one or two analyses in order to give some hints on this topic.

Reviewer #2 (Remarks to the Author):

The manuscript by Tremel et al describes the characterisation of the complex II vps34 lipid kinase complex on membranes using a wide synergy of methods, including electron microscopy, HDX-MS, MitoID, and careful biochemical and biophysical analysis. The authors have thoroughly investigated the molecular mechanisms by which Rab GTPases can activate lipid kinase activity through membrane recruitment. They have found an unexpected direct activation of class III PI3K by the GTPase Rab1, which preferentially activates only complex I. Using a mitoID approach they identified clearly that Rab1 only recruits class I and class II in a cellular context. As one of the main GEFs responsible for GTP loading Rab1 (TRAPPIII) has been linked to autophagy through a not fully understood mechanism, this is very exciting data.

This paper represents a major advance in understanding the role of Rabs in promoting autophagy through vps34, as well as a great expansion on the mechanisms by which Rab GTPases can recruit downstream effectors on membranes. It also is a beautiful use of electron microscopy to study a peripheral membrane complex in its native membrane environment. I am very positive on the manuscript, and think it should be published, and I have only minor corrections that should be possible without further experimentation.

Major

1. One of the most important discoveries in this manuscript is the unexpected role of Rab1 in activating only class III PI3K complex I. While Rab1 appears to bind in the same interface, there are

obviously critical differences between complex I and complex II in allowing for Rab1 activation. Can the authors address the following points

- a) Can the authors speculate on any differences in dynamics between complex I and complex II in the Rab binding region that allow for the complex specific Rab1 activation?
- b) Specifically is there any information that Atg14 is playing a role in binding, or is this solely through an altered conformation of vps34/vps15/beclin in complex I?
- c) There does not appear to be any regions in atg14 that are protected in HDX-MS experiments in Fig2e, but it might be worth showing regions with no coverage in a different color, as that might show if there are gaps in Atg14 coverage that may be why no difference is seen. Is the difference due to incomplete coverage or truly no difference.
- d) Also it may be useful to comment on any difference in C2 dynamics that are different between complex I and complex II from the authors previous HDX-MS analysis?

2. It might be useful to show an alignment of the class III PI3K residues in the putative Rab binding interface for Rab1 and Rab5 (ie the expected residues in ATG14 and UVRAG). Are there obvious differences in the Rab switches that can possibly account for differences, even in the low resolution structural data bound to Rab5?

3. One of the interesting observations from the Hurley lab in the paper Stjepanovic et al 2017 in Molecular Cell was that there was a large degree of dynamic movement of the kinase domain relative to the rest of the complex in their negative stain EM. Is this ever seen in EM/ET images of complex II? It seems from the images shown in Fig.4d that release of the kinase domain would seem to be a plausible mechanism for increasing activity once it is tethered to the membrane through the aromatic finger on Beclin I.

Can the authors speculate on this mechanism based on their own EM data.

4. The generation of the REIE mutant that toggles the selectivity for Rabs for complex I and II (increased for complex II with Rab5, and decreased for complex I with Rab1) seems to be a very important finding of this manuscript. This particularly could be a useful tool compound for many researchers. I worry that showing this in the Extended data section will make this harder to find for a broad audience, and would recommend moving this to a main figure, possibly combining Ext. Fig 2+3 into a new main text fig.

Minor typos/edits:

1. Line 119: typo: Surprisingly, Rab5a (the) colocalization was dramatically increased with complex II carrying mutated VPS34 REIE>AAAA (Fig. 1g).
2. HDX-MS data should include a supplemental table and description as recommended in Masson et al Nature Methods.

Reviewer #3 (Remarks to the Author):

This manuscript reveals how the VPS34 complex II is recruited to membranes and activated by Rab5a, based on cryo electron tomography of VPS34-Rab5a reconstituted in vesicles and complementary biophysical approaches. The manuscript is well written and the study is technically sound. The results will be of broad interest to researchers in the fields of intracellular membrane trafficking and autophagy and I recommend publication, after considering and addressing following points:

- The authors do not explain sufficiently, the reason why subtomogram-averaging was preferred for the visualization of the complex. An alternative workflow would be to reconstitute the complex into lipid nanodiscs, which would in turn allow collection of larger SPA datasets, extensive classifications

and elucidation of the different states, possibly at higher resolution. I suspect a possible reason might be membrane curvature. The authors indeed indicate in a recent study (Ohashi et al., eLife, 2020) that membrane curvature can greatly increase VPS34 activity. Most possibly, this is also the reason why the authors included in the classification only particles from vesicles of similar diameters and therefore state that “the motions of the complex with respect to the membrane are not likely to be caused by different membrane curvatures”. This is a crucial point and I encourage the authors to provide more background information and further discuss this in the revised manuscript. Furthermore, the sample contains different sizes of vesicles (Fig. 3a) and most importantly, vesicles can be flattened on the EM grid. The authors should further elaborate their criteria for selection of appropriate vesicles, that were then finally included in the analysis, to ensure that the different conformational states shown, indeed do not result from such differences.

- The authors indicate that the respective density was of sufficient resolution to unambiguously fit the crystal structure of Rab5a. This is not obvious however in Figure 3c. The authors should provide an additional short movie to demonstrate the quality of the fitting (the maps and models were not provided as part of the submission).
- Rab5a is linked to its TMD via a flexible linker peptide. Does the distance of the respective Term of Rab5a to the membrane (as fitted into the cryoEM volume), correlate well with the maximum length of this linker peptide (in the different classes)?.
- It would be good to discuss possible conformational changes and rearrangements, compared to the previous yeast VPS34 high resolution structure, in greater detail. Furthermore, albeit the moderate resolution of the cryoEM volume, the authors should be able to provide more details about the interface (for example, discuss the conservation of the suggested Rab5a interface and/or discuss mutants and their respective topology (in case data are available).
- I strongly encourage the authors to include brief descriptions of all methods carried out in this study and not only link to the descriptions provided previously (see LUV preparation, Flotation assay etc..)

Reviewer #4 (Remarks to the Author):

The paper by Tremel et al. provides important new information on the structure of Vps34 Complex II associated with membrane-bound Rab5, as well as the unexpected finding that Complex I binds not to Rab5 but to Rab1. The crosslinking and structural studies present the clearest analysis to date of how Vps34 functions on membranes.

The major problem with the paper is the use of the membrane-tethered Rab5 assay to demonstrate activation of the Vps34 kinase. Targeting of PI3Ks to membranes is well known to increase net lipid kinase activity, independent of kinase activation. This has been shown using CAAX-tagged Class I kinases, and has in fact been shown by Williams and Falcke for the Ras/PI3Ka interaction, where a translocation-induced increase in PIP3 production occurs despite the fact that the enzyme is actually inhibited. Thus, the fact that phosphorylation of PI is increased when Vps34 is bound to membrane-tethered Rab5 is not surprising, and does not demonstrate that Rab5 binding activates the enzyme.

Given the differences between the Complex II crystal structure versus the structure of membrane/Rab5-bound Complex II, it is certainly plausible that Rab5 binding increases the specific activity of the enzyme. However, membrane tethering drives other interactions as well, for example the association of the Beclin Bara domain with the membrane. It is not immediately clear how either the Beclin-membrane or Rab5-Vps15 interactions would drive the altered arrangement of the Vps15 N-lobe and the Vps34 active site.

The issue of Rab5 activation could be addressed by using a Rab5-independent method of targeting Complex II to membrane; this would presumably also increase PI(3)P production, but perhaps not as much as by membrane-tethered Rab5. Alternatively, soluble Rab5 (which was used by Zerial to first identify Vps34 as a Rab5 binding protein) could be added to the membrane tethered complex to see if additional activation occurs.

This reviewer acknowledges that that this would be a lot of work, and the data in the paper as it stands is important and should be published. Therefore, an editorial alternative would be to modify the title and text to indicate that the structure of Complex II bound to membrane-tethered Rab5 is consistent with Rab5-mediated activation, but to acknowledge that the increased activity in the lipid kinase assay does not definitively demonstrate this activation.

Additional points.

1. Fig. 1c. The GTP-dependence of Complex II-Rab5 binding in the flotation assay is not very robust – the data should be quantitated to show the increase in binding. Binding of Complex I should also be measured – this is important in order to show that the failure of Rab5 to increase Complex I activity in the assay is simply due to reduced membrane recruitment.

2. The microscopy data on localization of Vps34-EGFP with mCherry-Rab5 needs to be analyzed quantitatively, using Pearson's coefficient or some other measure of colocalization. Colocalization should be measured in more than one field of cells.

3. Fig. 3b. As in Fig. 1c, the reciprocal experiment with Complex II should be performed, to determine whether the low level of Complex II activity in the presence of Rab1 vesicles is due to reduced recruitment.

4. The colocalization in Fig. 2f is less convincing than in Fig 1, as localization of the Rab1a construct is less clean. Quantitative data from more than one field of cells should be provided. The data in 2f is also difficult to evaluate, as the Rab1a expression is greatly reduced.

Minor points.

1. E202 is difficult to see in the model in Fig. 1e.

Response to reviewers

We appreciate the many useful suggestions offered by the reviewers, and we are gratified that the reviewers are all enthusiastic about publication in Nature Communications. In the following, we respond to each of the concerns raised by the reviewers. The reviewer comments are in bold, and our response follows each point in normal type.

Reviewer 1

The manuscript by Tremel et al. investigates the structural basis of VPS34 activation on membranes. The authors established that Rab5a is a potent activator of the heterotetrameric core complex II containing VPS34 and mapped the Rab5a-GTP binding site on complex II using crosslinking and hydrogen deuterium exchange mass spectrometry. Also, expanding MitoID analysis to include Rab1 the authors detected all four subunit of the VPS34 complex 1 and further demonstrated that Rab1 interacts and activates VPS34 complex I. Furthermore, the authors did Cryo-ET to establish the structure of VPS34 complex II on RAB5-GTP positive vesicles and elucidated the key steps of the activation of VPS34 complex II by Rab5.

The manuscript is clearly written and presents a number of novel and interesting data mainly on the interaction between Rab5 and VPS34 complex II and its activation. Experiments are carefully performed with the appropriate controls and conclusions are based on solid and convincing results.

Thus, I believe that this manuscript deserves publication in Nature Communications although I have one general criticism regarding the part of the manuscript dealing with Rab1 that should be addressed before publication. While the authors go very deep in the definition of the steps regarding recruitment, binding and activation of VPS34 complex II by Rab5, the interesting results obtained on Rab1 and VPS34 complex I are somehow not pursued. Indeed, the results show clearly that Rab1 interacts and activates the autophagy associated VPS34 complex I, but no experiments are shown regarding the structural basis of this interaction and activation. It is clear that the authors cannot do all the experiments done on Rab5 during the short time of a revision but they could try to perform just one or two analyses in order to give some hints on this topic.

We appreciate that Reviewer 1 believes that the manuscript deserves publication in Nature Communications. The reviewer has raised the concern that there are no experiments showing the structural basis for Rab1a interacting with and activating complex I. While it is true that we have done far more with Rab5a, we have actually structurally characterised the Rab1a/complex I interaction, using HDX-MS.

We have carried out HDX-MS to examine the interaction of complex I with Rab1a (as shown in Fig. 3e). To prepare Fig. 3e, we made an effort to build and refine a model for complex I based on a report of a cryo-EM reconstruction of complex I bound to NRBF2 (EMD-20390). This is the only cryo-EM reconstruction that has been reported for complex I in which both arms of the complex are evident in the density, however, no atomic model of complex I has been deposited. We now have revised the manuscript to explain our building and refinement of a complex I model (see model building in the Methods section). By building the best model that we could for the cryo-EM density and refining it, we derived the basis for a structural interpretation of our results for differences in HDX caused by binding of Rab1a to

complex I. While we are keen to determine a structure of Rab1a bound to complex I, this is ongoing work in our lab, and it is beyond the scope of the current manuscript.

Our HDX-MS-based structural model strongly suggests that the Rab1a interacts with the same elements in complex I as Rab5a does for complex II. Indeed, we report the effect of a site-specific mutant of the VPS34 C2 domain that eliminates this interaction and eliminates Rab1a-mediated activation both in vitro and in cells.

Reviewer 2

The manuscript by Tremel et al describes the characterisation of the complex II vps34 lipid kinase complex on membranes using a wide synergy of methods, including electron microscopy, HDX-MS, MitoID, and careful biochemical and biophysical analysis. The authors have thoroughly investigated the molecular mechanisms by which Rab GTPases can activate lipid kinase activity through membrane recruitment. They have found an unexpected direct activation of class III PI3K by the GTPase Rab1, which preferentially activates only complex I. Using a mitoID approach they identified clearly that Rab1 only recruits class I and class II in a cellular context. As one of the main GEFs responsible for GTP loading Rab1 (TRAPPIII) has been linked to autophagy through a not fully understood mechanism, this is very exciting data.

This paper represents a major advance in understanding the role of Rabs in promoting autophagy through vps34, as well as a great expansion on the mechanisms by which Rab GTPases can recruit downstream effectors on membranes. It also is a beautiful use of electron microscopy to study a peripheral membrane complex in its native membrane environment. I am very positive on the manuscript, and think it should be published, and I have only minor corrections that should be possible without further experimentation.

We are pleased that Reviewer 2 is very positive about the manuscript and acknowledges the panoply of approaches that we have employed to understand the structural basis of activation of VPS34 complexes by Rabs. In particular, the reviewer recognises that the manuscript presents a “...beautiful use of electron microscopy to study a peripheral membrane complex in its native environment.” The reviewer suggests several changes.

1. One of the most important discoveries in this manuscript is the unexpected role of Rab1 in activating only class III PI3K complex I. While Rab1 appears to bind in the same interface, there are obviously critical differences between complex I and complex II in allowing for Rab1 activation. Can the authors address the following points

a) Can the authors speculate on any differences in dynamics between complex I and complex II in the Rab binding region that allow for the complex specific Rab1 activation?

We built and refined models of both complex I and complex II (see above). Using these models, it is clear that complex I has a distinct bend in its adaptor arm relative to complex II (Supplementary Fig. 6). The WD40/BARA2/BARA unit of complex II curls toward Rab5a binding site, whereas the analogous unit of complex I is positioned more distantly from the Rab-binding pocket. Consistent with this tighter packing in complex II, the VPS15 WD40

domain of complex II has several regions that show HDX changes in the presence of Rab5a, whereas no regions of the VPS15 WD40 in complex I are altered by Rab1a binding.

The helical insertion in the C2 domain of VPS34 (C2HH) tracks the movement of the VPS15 WD40 domain. As a consequence, the C2HH is tilted in complex I relative to complex II (Supplementary Fig. 6b). While we have now summarised these observations in our discussion on p. 11-12, it will be necessary to obtain higher resolution structural information in the future to fully understand the determinants of Rab specificity for the two complexes.

b. Specifically is there any information that Atg14 is playing a role in binding, or is this solely through an altered conformation of vps34/vps15/beclin in complex I?"

On the basis of HDX-MS observations, there is no observed peptide of ATG14L that appears to make a direct interaction with Rab1a. Although several regions of UVRAG have decreased HDX in the presence of Rab5a, it is not clear whether this is due to a direct interaction or due to the adaptor arm assuming a more compact conformation in the presence of Rab5a. From the structures available at this time, we have no evidence that either ATG14L or UVRAG make direct interactions with bound Rabs, but we cannot exclude that possibility. We have briefly summarised these observations in the Discussion on p. 11-12.

c. "There does not appear to be any region in atg14 that is protected in HDX-MS experiments in Fig2e, but it might be worth showing regions with no coverage in a different color, as that might show if there are gaps in Atg14 coverage that may be why no difference is seen, i.e., is the difference due to incomplete coverage or truly no difference. "

In order to avoid complicating the HDX-MS illustrations in Figs. 1 and 3, we have prepared a Supplementary Fig. 3 illustrating the distribution of unobserved regions in the HDX-MS results. There are unobserved regions, including a fairly long stretch in the coiled-coil region of ATG14L. In the discussion on p. 12, we have raised the possibility of direct interactions with regions unobserved in the HDX-MS experiments and lost in low-resolution structural interpretations.

d) Also it may be useful to comment on any difference in C2 dynamics that are different between complex I and complex II from the authors previous HDX-MS analysis?

The helical insertion in the C2 domain, which contacts Rab5a near VPS34 residue E202, shows HDX protection in both complex II in the presence of Rab5a and complex I in the presence of Rab1a. However, consistent with the concept that the WD40 of VPS15 is shifted away from the Rab1a binding pocket in complex I compared to the Rab5a pocket in complex II (see 1a above), there is generally less protection of the WD40 domain of complex I brought about by Rab1a binding. We have addressed this point in our revised discussion on p. 11.

2. It might be useful to show an alignment of the class III PI3K residues in the putative Rab binding interface for Rab1 and Rab5 (ie the expected residues in ATG14 and UVRAG). Are there obvious differences in the Rab switches that can possibly account for differences, even in the low resolution structural data bound to Rab5?

Given that there are no high-resolution structures of either human complex I or complex II, speculating on exact residues in ATG14L/UVRAG that could contribute to Rab specificity

would be an over-interpretation (especially since we have no supportive evidence that either makes a direct interaction with the Rabs). On the Rab side, there are potentially interesting differences between Rab1a and Rab5a in the regions of the interactions, but we believe they are too hypothetical to be included at this stage.

3. One of the interesting observations from the Hurley lab in the paper Stjepanovic et al 2017 in Molecular Cell was that there was a large degree of dynamic movement of the kinase domain relative to the rest of the complex in their negative stain EM. Is this ever seen in EM/ET images of complex II? It seems from the images shown in Fig.4d that release of the kinase domain would seem to be a plausible mechanism for increasing activity once it is tethered to the membrane through the aromatic finger on Beclin I. Can the authors speculate on this mechanism based on their own EM data.

Although our structures for the complex II classes have limited resolution, the same catalytic arm can be readily fit to the density for each of the well-defined classes shown in Fig. 6 and Supplementary Fig. 4, suggesting that there is no dramatic dislodging of the entire VPS34 HELCAT from the rest of the complex II, as described in Stjepanovic et al for complex I. We see only the opening between the VPS15 kinase domain and the VPS34 kinase domain. We have commented on this in the legend to Fig. 6 where we have shown the model fit to the density for each of the classes.

4. The generation of the REIE mutant that toggles the selectivity for Rabs for complex I and II (increased for complex II with Rab5, and decreased for complex I with Rab1) seems to be a very important finding of this manuscript. This particularly could be a useful tool compound for many researcher. I worry that showing this in the Extended data section will make this harder to find for a broad audience, and would recommend moving this to a main figure, possibly combining Ext. Fig 2+3 into a new main text fig.

We agree with the reviewer, and all of the results with the mutant activities have now been moved into the main Fig. 2 (for Rab5a) and Fig. 4 (for Rab1a).

Minor typos/edits:

1. Line 119: typo: Surprisingly, Rab5a (the) colocalization was dramatically increased with complex II carrying mutated VPS34 REIE>AAA (Fig. 1g).

Done

2. HDX-MS data should include a supplemental table and description as recommended in Masson et al Nature Methods.

Our HDX-MS results are presented in Excel spreadsheets. The journal's editorial manager made an unsuccessful attempt to convert these to PDF files. This was not our intention. The reviewer can see the Excel files by downloading the Reviewer zip file. We will take care that if these PDF files appear again, we will ask the administrator to remove them.

Reviewer #3

This manuscript reveals how the VPS34 complex II is recruited to membranes and activated by Rab5a, based on cryo electron tomography of VPS34-Rab5a reconstituted in vesicles and complementary biophysical approaches. The manuscript is well written and the study is technically sound. The results will be of broad interest to researchers in the fields of intracellular membrane trafficking and autophagy and I recommend publication, after considering and addressing following points:

The authors do not explain sufficiently, the reason why subtomogram-averaging was preferred for the visualization of the complex. An alternative workflow would be to reconstitute the complex into lipid nanodiscs, which would in turn allow collection of larger SPA datasets, extensive classifications and elucidation of the different states, possibly at higher resolution. I suspect a possible reason might be membrane curvature. The authors indeed indicate in a recent study (Ohashi et al., eLife, 2020) that membrane curvature can greatly increase VPS34 activity.

The reviewer is correct that our principle reason for using vesicles was the sensitivity of the enzyme complex to membrane curvature. We wanted to be able to investigate the influence of membrane curvature on the protein structure. In the end, we do not see any obvious pattern among the vesicles that we have used for the cryo-ET, although the range of curvatures represented in the data set is limited.

We appreciate the reviewer's suggestion, and we have started to explore this nanodisc strategy for other complexes. Nanodiscs have their own difficulties, and we have not yet been successful with this approach for any peripheral complex. Indeed we know of no structure of a peripheral membrane protein complex that has been reported using nanodiscs. Part of the difficulty for these complexes might be that they do not have very high affinity interactions with membranes, and this transience is essential for their biological regulation. To get nanodiscs with a high occupancy might be possible only by saturating the system with complexes so that the grid will have a very large population of unbound complexes. This is true for the cryo-ET of vesicles as well, but we are able to use the vesicle as a filter so that unbound complexes do not interfere.

Most possibly, this is also the reason why the authors included in the classification only particles from vesicles of similar diameters and therefore state that “the motions of the complex with respect to the membrane are not likely to be caused by different membrane curvatures”. This is a crucial point and I encourage the authors to provide more background information and further discuss this in the revised manuscript.

Furthermore, the sample contains different sizes of vesicles (Fig. 3a) and most importantly, vesicles can be flattened on the EM grid. The authors should further elaborate their criteria for selection of appropriate vesicles, that were then finally included in the analysis, to ensure that the different conformational states shown, indeed do not result from such differences.

We believe there may also be a misunderstanding, and we have sought to clarify this. Vesicles of a wide range of diameters were chosen (from ~60 nm to ~150 nm, as seen in the whisker plot shown in Supplementary Fig. 4b). Only vesicles which had invaginations or were very deformed were skipped in picking (we have added this comment to the methods). All of these vesicles were included in classification. The same mean broad distribution of

vesicle diameters is maintained in each class, hence the movement on the membrane surface is independent of curvature for this population of vesicles. We have stated this more clearly in the legend to supplementary figure 4: *“Classes 1-3 from the 3D classification have different orientations relative to the membrane (see panel c), but have similar vesicle diameter distributions, suggesting that different orientations of the complex II with respect to the membrane are not caused by different membrane curvatures.”*

There is some flattening of the vesicles, but we did not make use of particles at the top and bottom of these vesicles because they contact or protrude through the air-water interface. We only use the regions near the equator of the vesicles for analysis.

The authors indicate that the respective density was of sufficient resolution to unambiguously fit the crystal structure of Rab5a. This is not obvious however in Figure 3c. The authors should provide an additional short movie to demonstrate the quality of the fitting (the maps and models were not provided as part of the submission).

We have included a movie in the supplemental data that illustrates the fit of the Rab5a (orange) into the density, as shown below. This is now Supplementary Movie 1.

-Rab5a is linked to its TMD via a flexible linker peptide. Does the distance of the respective Term of Rab5a to the membrane (as fitted into the cryoEM volume), correlate well with the maximum length of this linker peptide (in the different classes)?

The length of the flexible hypervariable region is ample to span the Rab5a-binding pocket and the membrane. We have added to the legend of Fig. 6b, “The distance between the Rab5a density and membrane is approximately 50 Å, which easily can be spanned by the 34 residues forming the Rab5a hypervariable region (not ordered in the structure).”

It would be good to discuss possible conformational changes and rearrangements, compared to the previous yeast VPS34 high resolution structure, in greater detail. Furthermore, albeit the moderate resolution of the cryoEM volume, the authors should be able to provide more details about the interface (for example, discuss the conservation of the suggested Rab5a interface and/or discuss mutants and their respective topology (in case data are available).

We greatly appreciate the enthusiasm of the reviewer for this point that we certainly share. However, the subunits of the yeast complex II are distant orthologs from those of human complex II. For example VPS15, which is the scaffold of complex II has only about 25% sequence identity with the human VPS15. While the fold is immediately recognisable, not every loop is ordered. Since at this resolution there are no side-chains visible, our identification of a given residue is based on a structural superposition of the human and yeast models. Because of this, we believe that a detailed analysis of global alignments with the yeast structure is probably not helpful at this stage. However, because we now have similar resolution models for human complexes I and II (see response to reviewer I), a structural comparison of human complex I and yeast complex II is more interesting, because it can highlight features that could account for the Rab selectivities of the two complexes. We find that there is a distinct shift of the WD40 domain of VPS15 that shifts the VPS34 C2 domain helical insertion, which forms all of the interactions with the switch regions of Rab5a. This Rab specificity could be accomplished with no direct contact between the Rabs and the complex-specific subunits ATG14L and UVRAG, which is suggested by the cryo-EM structures and the HDX-MS, although at this resolution we cannot rule out direct, transient interactions. We have addressed these observations in our response to point 1a of Reviewer #2. We have added a new figure (Supplementary Fig. 6b) and revised the Discussion on p. 11-12.

The primary contact between Rab5a and complex II is the first helix of the VPS34 C2 domain helical insertion. This is likely to be true for Rab1a. This suggests that a stretch of no more than 20 residues (190-210) in VPS34 complex II imparts the nucleotide-specific recognition essential to the role of Rab5a in activating complex II in cells. This region in VPS34 is completely conserved in nearly all VPS34 orthologues in animals and plants, but there are variations in yeast that might point to variations in the Rab5a orthologue of yeast. There are differences in the surfaces of Rab1a and Rab5a but they are subtle, and we believe that it will require higher resolution structures to fully understand the determinants of Rab-selectivity of the VPS34 complexes. Because of this, we believe that more detailed discussion at this stage is too speculative and beyond the scope of the manuscript.

I strongly encourage the authors to include brief descriptions of all methods carried out in this study and not only link to the descriptions provided previously (see LUV preparation, Flotation assay etc..)

We have expanded on details of the methods in the main text.

Reviewer #4

The paper by Tremel et al. provides important new information on the structure of Vps34 complex II associated with membrane-bound Rab5, as well as the unexpected finding that complex I binds not to Rab5 but to Rab1. The crosslinking and structural studies present the clearest analysis to date of how Vps34 functions on membranes.

We thank the reviewer for the positive view of the manuscript.

The major problem with the paper is the use of the membrane-tethered Rab5 assay to demonstrate activation of the Vps34 kinase.

Lipidation of Rab5 is the biological manner in which Rab5 is targeted to membranes. Our method of lipidation is somewhat different to the isoprenylation that is carried out in cells, but membrane tethering is true of both our in vitro system and the cellular system.

Targeting of PI3Ks to membranes is well known to increase net lipid kinase activity, independent of kinase activation. This has been shown using CAAX-tagged Class I kinases, and has in fact been shown by Williams and Falcke for the Ras/PI3Ka interaction, where a translocation-induced increase in PIP3 production occurs despite the fact that the enzyme is actually inhibited. Thus, the fact that phosphorylation of PI is increased when Vps34 is bound to membrane-tethered Rab5 is not surprising, and does not demonstrate that Rab5 binding activates the enzyme.

In the work with Falcke for a class IA PI3K, we used a lipidated Ras, an approach very closely related to what we have done in the current manuscript for Rab1a and Rab5a. The reviewer has correctly noted that in our work with the class I PI3K recruited to Ras on a membrane, the PI3K had a lower specific activity than the same PI3K on a membrane without Ras. However, because the density of PI3K is greater on membranes with lipidated Ras, the Ras resulted in a net increase in PI3K activity.

Thus, the fact that phosphorylation of PI is increased when Vps34 is bound to membrane-tethered Rab5 is not surprising, and does not demonstrate that Rab5 binding activates the enzyme.

Having Rab5 on the membrane is the biological state in cells, and our work shows that this state leads to greater VPS34 activity in our reconstituted system. However, the reviewer is correct that this does not answer the question whether the specific activity of complex II bound to membranes with Rab5a is greater than the specific activity of the complex bound to membranes in the absence of Rab5a. We agree with the reviewer that this mechanistic question is important, and we appreciate his astute recognition of it. Consequently, we are happy to say that this question was recently answered (during the review of this paper) by the elegant study of Joseph Falcke in which we had the pleasure to collaborate (Buckles et al. Biophys J. Oct 2020, PMID 33137306). This single molecule kinetic study on supported lipid bilayers showed that Rab5a has two effects: (1) It increases the density of complex II on membranes, by recruiting complex II to membranes, and (2) every complex II on the membrane in the presence of Rab5a has a greater specific activity than it has on membranes in the absence of Rab5a. This result dovetails nicely with our structural observations that complex II on membranes with lipidated Rab5a has a different conformation than complex II in the absence of membranes (VPS34 loses the inhibitory contact with VPS15). We have modified our Discussion in light of the Falcke results, and we have cited the Buckles et al paper (reference 53).

Given the differences between the complex II crystal structure versus the structure of membrane/Rab5-bound complex II, it is certainly plausible that Rab5 binding increases the specific activity of the enzyme. However, membrane tethering drives other interactions as well, for example the association of the Beclin Bara domain with the membrane. It is not immediately clear how either the Beclin-membrane or Rab5-Vps15 interactions would drive the altered arrangement of the Vps15 N-lobe and the Vps34 active site.

The reviewer is correct that we only know that upon binding to Rab5a on membranes, complex II has an increased specific activity, and that the auto-inhibitory interaction of VPS34 with the VPS15 kinase domain is lost. The mechanism by which this allosteric activation is brought about is not immediately obvious.

We do know that soluble Rab5a has no influence on complex II activity (Supplementary Figs. 1b,c), whereas Rab5a coupled to membranes greatly activates complex II. This suggests that the allosteric change in conformation that we see in the structure may be a direct consequence of increased association of complex II with membranes. The structure shows that the adaptor arm is in contact with the membrane whereas the kinase domain is not. This could mean that the catalytic arm undergoes multiple micro hops on and off the membrane before the adaptor arm releases, and that the state with both arms in contact with membranes is only a small part of the catalytic cycle (we suggested this in the Discussion on p. 11). The TIRFM approach, which was employed in our collaboration paper with the Falke lab (PMID 33137306) would only detect when the whole complex leaves the membrane. It may mean that by Rab5a holding the complex on the membrane longer, there are more micro hops, which would increase the likelihood of the inactive-to-active transition. This mechanism remains speculative, and conclusively testing this speculation would require new experimental approaches that we believe would be beyond the scope of this manuscript.

The issue of Rab5 activation could be addressed by using a Rab5-independent method of targeting complex II to membrane; this would presumably also increase PI(3)P production, but perhaps not as much as by membrane-tethered Rab5. Alternatively, soluble Rab5 (which was used by Zerial to first identify Vps34 as a Rab5 binding protein) could be added to the membrane tethered complex to see if additional activation occurs.

The reviewer suggests a possible strategy to see if Rab5a increases specific activity of complex II. However, the work of Falke and his colleagues described above has now answered this question.

This reviewer acknowledges that that this would be a lot of work, and the data in the paper as it stands is important and should be published.

We are pleased that the reviewer recognises the strengths and importance our work.

Therefore, an editorial alternative would be to modify the title and text to indicate that the structure of complex II bound to membrane-tethered Rab5 is consistent with Rab5-mediated activation, but to acknowledge that the increased activity in the lipid kinase assay does not definitively demonstrate this activation.

In our recent study with Falke and colleagues (Buckles et al., 2020, ref 53), we have unequivocally established that there is an increase in specific activity of complex II on membranes with Rab5a over the complex on membranes in the absence of Rab5a. We now cite this in our Discussion on p. 10. We know that there is a structural change eliminating an autoinhibitory contact with VPS15, and we know that complex II has a greater specific activity on Rab5a membranes. Consequently, we believe the title of the manuscript is now fully justified.

Additional points.

1. Fig. 1c. The GTP-dependence of complex II-Rab5 binding in the flotation assay is not very robust – the data should be quantitated to show the increase in binding. Binding of complex I should also be measured – this is important in order to show that the failure of Rab5 to increase complex I activity in the assay is simply due to reduced membrane recruitment.

These are now quantitated in Supplementary Fig. 2. We also comment on p. 3 that the failure of Rab5a to activate complex I to the same extent as complex II is likely to be due to Rab5a having less impact on the affinity of complex I for membranes.

2. The microscopy data on localization of Vps34-EGFP with mCherry-Rab5 needs to be analyzed quantitatively, using Pearson's coefficient or some other measure of colocalization. Colocalization should be measured in more than one field of cells.

The co-localisations are now quantitated (as described in the methods p. 27) and are shown in Figs. 2e and 4e. In addition, we show the fluorescence traces to the right of each micrograph in new Figs. 2a-d and 4a-d.

3. Fig. 3b. As in Fig. 1c, the reciprocal experiment with complex II should be performed, to determine whether the low level of complex II activity in the presence of Rab1 vesicles is due to reduced recruitment.

This is now quantitated in Supplementary Fig. 2. We also comment on p. 3 that the failure of Rab1a to activate complex II to the same extent as complex I is likely to be due to Rab1a having less impact on the affinity of complex II for membranes.

4. The colocalization in Fig. 2f is less convincing than in Fig 1, as localization of the Rab1a construct is less clean. Quantitative data from more than one field of cells should be provided. The data in 2f is also difficult to evaluate, as the Rab1a expression is greatly reduced.

In new Figs. 2 and 4 we have provided plot profiles of EGFP/mCherry channels for transects across the selected cells. Furthermore, the quantification of the recruitment of complexes I and II to the Rab1a compartment has been provided in Fig. 4e that was in each case obtained from more than three fields of cells. We agree that in the previous micrographs (old Figure 2g), the Rab1a expression appeared to be greatly reduced. However, this was simply due to a technical issue of image display (insufficient brightness). We have chosen more representative micrographs that contain more cells, and adjusted the brightness of these micrographs. We believe these results are quantitatively convincing to show the effect of the VPS34 C2HH mutation in complex I on the recruitment to the Rab1a compartment.

Minor points.

1. E202 is difficult to see in the model in Fig. 1e.

An expanded view of the C2 helical insert now is shown in Fig. 1e.

REVIEWERS' COMMENTS

Reviewer #1 (Remarks to the Author):

The authors responded satisfactorily to my criticism thus I recommend publication.

Reviewer #2 (Remarks to the Author):

The authors have addressed all of my points, I recommend the version as written for publication

Reviewer #4 (Remarks to the Author):

The authors have addressed all my concerns, and the citation of the new Falke paper is an important addition.

Only one minor point - the quantitation of the vesicle flotation assay in Supp. Fig. 2 does not show error bars. If this is because the quantitation is of the single experiment shown to the left of each graph, then the figure legend should state how many times the experiment was performed.

Response to reviewers

We are gratified that the reviewers are all enthusiastic about publication in Nature Communications. The reviewer comments are in bold, and our response follows each point in normal type.

Reviewer 1

The authors responded satisfactorily to my criticism thus I recommend publication.

We appreciate the reviewer comment.

Reviewer 2

The authors have addressed all of my points, I recommend the version as written for publication

We appreciate the reviewer comment.

Reviewer #4

The authors have addressed all my concerns, and the citation of the new Falke paper is an important addition.

Only one minor point - the quantitation of the vesicle flotation assay in Supp. Fig. 2 does not show error bars. If this is because the quantitation is of the single experiment shown to the left of each graph, then the figure legend should state how many times the experiment was performed.

We thank the reviewer for the positive view of the manuscript. We have modified the text of supplementary Fig. 2 to answer the reviewer's question.